# Experimental quantum simulation of fermion-antifermion scattering via boson exchange in a trapped ion

Xiang Zhang[1,2], Kuan Zhang[1], Yangchao Shen[1], Shuaining Zhang[1], Jing-Ning Zhang[1], Man-Hong Yung[1,3,4], Jorge Casanova[5], Julen S. Pedernales [6], Lucas Lamata [6], Enrique Solano[6,7,8] & Kihwan Kim [1]

Quantum field theories describe a variety of fundamental phenomena in physics. However, their study often involves cumbersome numerical simulations. Quantum simulators, on the other hand, may outperform classical computational capacities due to their potential scalability. Here we report an experimental realization of a quantum simulation of fermion–antifermion scattering mediated by bosonic modes, using a multilevel trapped ion, which is a simplified model of fermion scattering in both perturbative and non-perturbative quantum electrodynamics. The simulated model exhibits prototypical features in quantum field theory including particle pair creation and annihilation, as well as self-energy interactions. These are experimentally observed by manipulating four internal levels of a $^{171}Yb^+$ trapped ion, where we encode the fermionic modes, and two motional degrees of freedom that simulate the bosonic modes. Our experiment establishes an avenue towards the efficient implementation of field modes, which may prove useful in studies of quantum field theories including non-perturbative regimes.

---

[1] Center for Quantum Information, Institute for Interdisciplinary Information Sciences, Tsinghua University, Beijing, 100084, China. [2] Department of Physics, Renmin University of China, Beijing, 100872, China. [3] Institute for Quantum Science and Engineering and Department of Physics, Southern University of Science and Technology of China, Shenzhen, 518055, China. [4] Shenzhen Key Laboratory of Quantum Science and Engineering, Shenzhen, 518055, China. [5] Institut für Theoretische Physik and IQST, Universität Ulm, Albert-Einstein-Allee 11, D-89069 Ulm, Germany. [6] Department of Physical Chemistry, University of the Basque Country UPV/EHU, Apartado 644, 48080 Bilbao, Spain. [7] IKERBASQUE, Basque Foundation for Science, Maria Diaz de Haro 3, 48013 Bilbao, Spain. [8] Department of Physics, Shanghai University, 200444 Shanghai, China. Xiang Zhang and Kuan Zhang contributed equally to this work. Correspondence and requests for materials should be addressed to J.-N.Z. (email: jnzhang13@mail.tsinghua.edu.cn) or to K.K. (email: kimkihwan@mail.tsinghua.edu.cn)

Quantum simulators are devices designed to predict the properties of physical models associated with target quantum systems[1, 2]. Their intrinsic physical behaviors are fully governed by the laws of quantum mechanics, making it possible to efficiently study complex quantum systems that cannot be solved by classical computers[3, 4]. Trapped ions and superconducting circuits have proved to be promising for experimentally simulating a variety of paradigmatic quantum models such as various spin models[5–9], relativistic Dirac equations[10–13], embedding quantum simulators[14–18], and fermionic models[19, 20]. More recently, a digital quantum simulation of a fermionic lattice gauge theory has been performed in trapped ions[21]. However, it would be desirable to realize a quantum simulator that involves interacting fermionic and bosonic fields as described by quantum field theories (QFTs)[22]. In this sense, fermionic modes could be mapped in the ion internal levels, whereas bosonic modes could be naturally encoded in the motional degrees of freedom.

Here we report an experimental quantum simulation of interacting fermionic and bosonic quantum field modes, where fermions are encoded in four internal levels of an $^{171}$Yb$^+$ ion and the bosonic modes in the motional degrees of freedom, following the proposal by Casanova et al.[23]. Therefore, this analog quantum simulation constitutes a step forward towards a digital-analog quantum simulator[19, 24–27] of perturbative and non-perturbative QFTs. In this sense, a remarkable feature of our experiment is that it contains all orders in perturbation theory, which is equivalent to all Feynman diagrams for a finite number of fermionic and bosonic modes. Moreover, our approach could in principle be scaled up by progressively adding more ions allowing the codification of additional fermionic and bosonic field modes, which may lead to full quantum simulations of QFTs such as quantum electrodynamics (QEDs)[22].

## Results

### Hamiltonian for quantum simulation of QFT

The common way to analyze QFTs is via a Dyson series expansion in perturbation theory and Feynman diagrams[22]. If we consider larger coupling parameters, standard perturbative methods become cumbersome for a finite-mode Dyson expansion, mainly because only a reduced number of perturbative Feynman diagrams can be calculated. On the other hand, a trapped-ion quantum simulator with its high degree of quantum control[28] could overcome these limitations and simulate QFTs more efficiently than classical computers[29]. Based on the proposal of ref.[23], our experimental quantum simulation of finite-number interacting quantized field modes includes all terms of the Dyson expansion. We experimentally implement a fundamental QFT model in a single trapped-ion considering (i) one fermionic and one antifermionic field mode, (ii) one or two bosonic field modes, which already reveals interesting QFT features such as self-interactions, particle creation and annihilation, and perturbative and non-perturbative regimes. The general Hamiltonian involving the continuum of fermionic and bosonic fields reads

$$H = \int dp\, \omega_p \left( \hat{b}_p^\dagger \hat{b}_p + \hat{d}_p^\dagger \hat{d}_p \right) + \int dk\, \omega_k \hat{a}_k^\dagger \hat{a}_k$$
$$+ g \int dx\, \psi^\dagger(x)\psi(x)A(x), \tag{1}$$

where $b_p$ and $d_p$ are fermionic and antifermionic annihilation operators, respectively, whereas $a_k$ are the bosonic annihilation operators. Here, $\omega$ ($\omega_k$) is the fermion and antifermion free energy (boson free energy), whereas $\psi(x)$ denotes the fermionic and $A(x)$ the bosonic fields[23].

As a stepped experimental demonstration, we first consider the simplest situation with only one bosonic mode, which can be

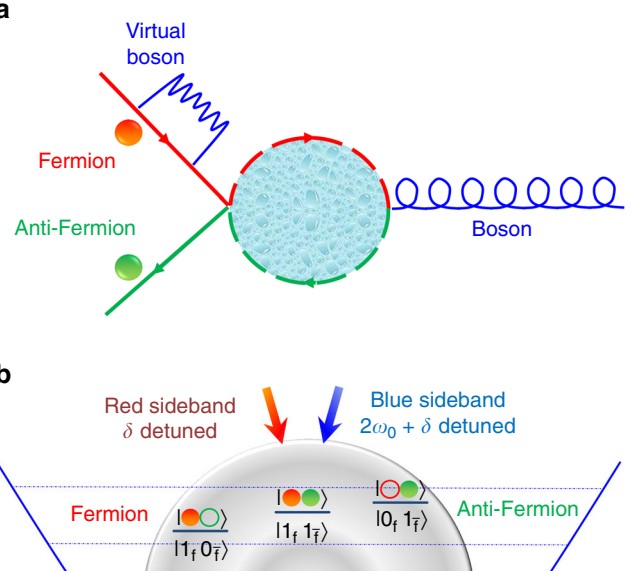

**Fig. 1** Fermion–antifermion scattering process and its mapping to an $^{171}$Yb$^+$ ion system. **a** Diagram of the interactions between a fermion, an antifermion, and bosons. The fermion emits and absorbs virtual bosons through the self-interaction process. In the fermion–antifermion scattering process, the middle dashed loop represents the summation of all terms in a finite-mode Dyson series expansion. **b** Diagram of the encoding and operations to implement the interaction Hamiltonian $H_I$ with an $^{171}$Yb$^+$ trapped ion. The vacuum state and the fermionic states are mapped onto four internal states through the Jordan–Wigner mapping. The bosonic mode is directly implemented with the vibrational mode along the $X$ radial direction. The self-interaction is implemented by a displacement operation, which shifts the center of the harmonic oscillator without changing the internal states. The fermion and antifermion scattering is simulated by the combination of the red- and the blue-sideband transitions, which change the internal states together with the vibrational mode

implemented by a single vibrational mode of the ion. The fermion and antifermion modes are considered as two comoving modes describing incoming Gaussian wave packets, which are centered in the average momentum and have distant average initial positions[23]. These modes describe self-interacting dressed states by emission and absorption of virtual bosons. They also represent the lowest-order in perturbation theory of the scattering of the incoming wave packets that will collide in a certain region of spacetime. The pair creation and annihilation is local and takes place only when the two wave packets of the fermion and antifermion overlap, namely, when the particles scatter. A diagram of these interactions, in the spirit of a Feynman diagram, is shown in Fig. 1(a). It is noteworthy that the loop of this diagram includes all terms in a finite-mode Dyson expansion, which is different from the standard perturbative approach with only a reduced number of Feynman diagrams. By considering slow massive bosons, as described in ref.[23], the interaction

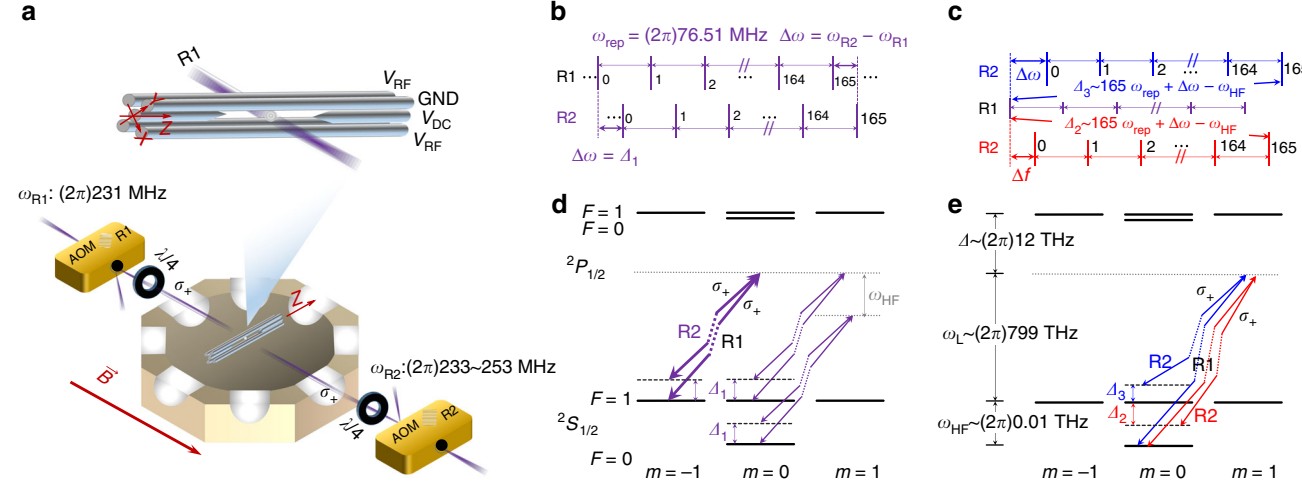

**Fig. 2** Schematic of the experimental implementation. **a** Experimental setup of the four-rod ion trap inside an octagon vacuum chamber and the geometry of Raman laser beams through the acousto-optic modulators (AOM R1 and R2). The two AOMs are driven with different frequencies $\omega_{R1}$ and $\omega_{R2}$, where $\omega_{R1}$ is fixed at $(2\pi)231$ MHz and $\omega_{R2}$ is adjusted in the range of $(2\pi)233 \sim 253$ MHz. Quarter-wave plates are used for polarization adjustment of the laser beams. **b**, **c** Frequency combs of the pico-second pulsed lasers and choice of the effective Raman beat-note frequency. The frequency interval of the "comb" is the repetition rate of the laser pulse, which is stabilized at $\omega_{rep} = (2\pi)76.51$ MHz. The frequency difference between the two AOMs is tuned near to the trap frequency $\omega_X$ (**b**) for the displacement operation or (**c**) to produce the hyperfine frequency with the addition of 165 intervals, $\omega_{HF}$. **d**, **e** The basic level structure and transitions of $^{171}$Yb$^+$ system coupled by $\sigma_+$ polarized Raman laser beams. The beat-note frequency of the Raman beams (**d**) for the displacement operation is $\Delta_1 = \omega_X - \omega_0$, where $\omega_0 = (2\pi)0.01$ MHz. Thick lines represent the two times stronger displacement operation on state $|1_f 0_{\bar{f}}\rangle$. **e** The frequency difference between the carrier transition and the red-sideband (blue sideband) operation is $\Delta_2 = \omega_X - \delta$ ($\Delta_3 = \omega_X - (2\omega_0 + \delta)$)

Hamiltonian we would like to realize turns into

$$H_I = g_1 e^{-i\omega_0 t}\left(\hat{b}^\dagger \hat{b}\hat{a}_0 + \hat{d}\hat{d}^\dagger \hat{a}_0\right)$$
$$+ g(t)\left(e^{i\delta t}\hat{b}^\dagger \hat{d}^\dagger \hat{a}_0 + e^{-i(2\omega_0+\delta)t}\hat{d}\hat{b}\hat{a}_0\right) \quad (2)$$
$$+ \text{H.c.},$$

where the associated time-dependent coupling strength is

$$g(t) = g_2 e^{-(t-T/2)^2/(2\sigma_t^2)}, \quad (3)$$

and $\delta = \omega_f + \omega_{\bar{f}} - \omega_0$. Here, $\omega_f$, $\omega_{\bar{f}}$, and $\omega_0$ represent the energy of the fermionic field mode $b$, the antifermionic field mode $d$, and the bosonic field mode $a_0$, respectively. The ratio $g_2/g_1$ gives the relative strength between pair creation and self-interaction. $T$ is the total time of the pair-creation process, whereas $\sigma_t$ is the temporal interval of the interaction region. Our formalism, explained in detail in ref.[23], involves considering incoming comoving fermionic and antifermionic modes at the lowest order in perturbation theory. The time dependence of the interaction of the incoming particles, as they collide, maps onto a time dependence of the interaction Hamiltonian coupling.

Applying a Jordan–Wigner mapping[23] from the fermionic modes to a $2 \times 2$ Hilbert space,

$$\hat{b}^\dagger = \hat{I} \otimes \hat{\sigma}^+, \hat{b} = \hat{I} \otimes \hat{\sigma}^-, \quad (4)$$

$$\hat{d}^\dagger = \hat{\sigma}^+ \otimes \hat{\sigma}_z, \hat{d} = \hat{\sigma}^- \otimes \hat{\sigma}_z, \quad (5)$$

the vacuum state and fermionic states are represented by

$$|\downarrow\downarrow\rangle = |0_f 0_{\bar{f}}\rangle, \qquad |\downarrow\uparrow\rangle = |1_f 0_{\bar{f}}\rangle,$$
$$|\uparrow\downarrow\rangle = -|0_f 1_{\bar{f}}\rangle, \quad |\uparrow\uparrow\rangle = -|1_f 1_{\bar{f}}\rangle, \quad (6)$$

where $|1_f 1_{\bar{f}}\rangle$ denotes the state containing one fermion and one

antifermion. Thus, the interaction Hamiltonian reads

$$H_I = g_1\left(|0_f 0_{\bar{f}}\rangle\langle 0_f 0_{\bar{f}}| + 2|1_f 0_{\bar{f}}\rangle\langle 1_f 0_{\bar{f}}|\right.$$
$$\left. + |1_f 1_{\bar{f}}\rangle\langle 1_f 1_{\bar{f}}|\right)\hat{a}_0 e^{-i\omega_0 t}$$
$$- g(t)\left(|0_f 0_{\bar{f}}\rangle\langle 1_f 1_{\bar{f}}|\hat{a}_0^\dagger e^{-i\delta t}\right.$$
$$\left. + |0_f 0_{\bar{f}}\rangle\langle 1_f 1_{\bar{f}}|\hat{a}_0 e^{-i(2\omega_0+\delta)t}\right) + \text{H.c.} \quad (7)$$

**Trapped ion implementation.** We point out that, due to the asymmetric role of fermionic annihilation and antifermionic creation operators in the fermionic field, the one-antifermion state is a dark state of the Hamiltonian in equation (2) and therefore the antifermion does not have self-energy at first order (it has, when considering more modes and higher orders). We implement the Hamiltonian on a single $^{171}$Yb$^+$ ion trapped in a three-dimensional harmonic potential as shown in Fig. 1(b). The radial harmonic potential is generated by an oscillating electric field $V_{RF}$ in the radial direction with the two trapping frequencies along $X$ and $Y$ directions being $(\omega_X, \omega_Y) = (2\pi)$ (2.4, 1.9) MHz. The bosonic modes are mapped onto these radial vibrational modes and we choose the $X$ mode for experiments involving a single bosonic mode. The vacuum state $|0_f 0_{\bar{f}}\rangle$ is mapped to the hyperfine state $|F = 0, m = 0\rangle$. Fermionic states are mapped onto Zeeman states as $|F = 1, m_F = -1\rangle \equiv |1_f 0_{\bar{f}}\rangle$, $|F = 1, m_F = 1\rangle \equiv |0_f 1_{\bar{f}}\rangle$, and $|F = 1, m_F = 0\rangle \equiv |1_f 1_{\bar{f}}\rangle$. With the mapping of the bosonic mode and the fermionic states onto the $^{171}$Yb$^+$ ion system, the Hamiltonian (7) is naturally divided into three parts: displacement, red-sideband, and blue-sideband operations.

The operations of the self-interaction and scattering processes of the fermion and the anti-fermion are realized by $\sigma_+$-polarized Raman laser beams[30–32] counter-propagating along the direction of the magnetic field $\vec{B}$. The strength of the magnetic field at the position of the ion is around 7 G, which produces $\omega_{ZM} = (2\pi)10$ MHz Zeeman splitting. The magnetic field is generated by a pair of Helmholtz coils and is aligned along the angle bisector

direction of the $X$ and $Y$ axes, which allows the laser beams to couple both of the vibrational modes, as shown in Fig. 2(a). The laser beams are modulated with acousto-optic modulators (AOMs), which are driven with different frequencies $\omega_{R1}$ and $\omega_{R2}$.

For the Raman transitions, the mode-locked picosecond laser is used with a wavelength of 375 nm, which is $\Delta = (2\pi)12$ THz red detuned from the optical transition $^2S_{1/2} \leftrightarrow {}^2P_{1/2}$. The train of laser pulses in the time domain can be considered as an equally spaced frequency "comb"[33], which in our experiment had a repetition rate of $\omega_{rep} = (2\pi)76.51$ MHz. As shown in Fig. 2(b, c), we use the frequency "comb" to select a Raman beat-note frequency according to the relation $\omega_R = \Delta\omega + n \times \omega_{rep}$, where $\Delta\omega = \omega_{R2} - \omega_{R1}$ and $n = 0, \pm 1, \pm 2, \ldots$. For transitions between different motional levels of the same electronic state, we simply use $n = 0$ and make $\Delta\omega$ close to $\omega_X$ or $\omega_Y$. For transitions between two different electronic states, we use $n = 165$.

Figure 2(d, e) shows the Raman schemes needed to implement Hamiltonian (7), which are naturally divided into three parts, namely,

$$g_1 \left( \left|0_f 0_{\bar{f}}\right\rangle\left\langle 0_f 0_{\bar{f}}\right| + 2\left|1_f 0_{\bar{f}}\right\rangle\left\langle 1_f 0_{\bar{f}}\right| \right. \\ \left. + \left|1_f 1_{\bar{f}}\right\rangle\left\langle 1_f 1_{\bar{f}}\right| \right) a_0 e^{-i\omega_0 t} \quad (8)$$

$$-g(t)\left|0_f 0_{\bar{f}}\right\rangle\left\langle 1_f 1_{\bar{f}}\right| a_0^\dagger e^{-i\delta t} \quad (9)$$

$$-g(t)\left|0_f 0_{\bar{f}}\right\rangle\left\langle 1_f 1_{\bar{f}}\right| a_0 e^{-i(2\omega_0 + \delta)t}. \quad (10)$$

Here, the first part is $\omega_0$-detuned displacement operation, the second part is $\delta$-detuned red-sideband operation between $\left|0_f 0_{\bar{f}}\right\rangle \leftrightarrow \left|1_f 1_{\bar{f}}\right\rangle$, and the last part is $(2\omega_0 + \delta)$-detuned blue-sideband operation between $\left|0_f 0_{\bar{f}}\right\rangle \leftrightarrow \left|1_f 1_{\bar{f}}\right\rangle$.

The first part corresponds to a displacement operation with $1 : 2 : 1 : 0$ relative ratios among the strength coefficients of states $\left|0_f 0_{\bar{f}}\right\rangle$, $\left|1_f 0_{\bar{f}}\right\rangle$, $\left|1_f 1_{\bar{f}}\right\rangle$, and $\left|0_f 1_{\bar{f}}\right\rangle$. Figure 2(d) shows how to implement the displacement operation through the counter-propagating Raman laser beams shown in Fig. 2(a). The $\sigma_+$-polarized Raman beams produce the exact ratios in the strength of displacement operations, as state $\left|1_f 0_{\bar{f}}\right\rangle$ is coupled to two levels in the $^2P_{1/2}$ manifold, states $\left|0_f 0_{\bar{f}}\right\rangle$, and $\left|1_f 1_{\bar{f}}\right\rangle$ to one level, and state $\left|0_f 1_{\bar{f}}\right\rangle$ to no level. The strength coefficient of a Raman path is given by $\Omega_R = g_1 g_2 / 2\Delta_R$, where $g_1$ and $g_2$ are Rabi frequencies of the two Raman beams coupled to the transition between $^2S_{1/2}$ and $^2P_{1/2}$ and the detuning $\Delta_R \approx \Delta = (2\pi)12$ THz. The coefficients of all possible Raman paths are added, as all optical transitions between $^2S_{1/2} \leftrightarrow {}^2P_{1/2}$ states have the same coefficients in absolute values. We note that the frequency difference $\omega_{HF} = (2\pi)12.6$ GHz between states $\left|0_f 0_{\bar{f}}\right\rangle$ and $\left|1_f 1_{\bar{f}}\right\rangle$ is much smaller than the detuning $\Delta$ of the Raman laser beams acting on the manifold $^2P_{1/2}$, which produces around a 0.1% difference in the strength of the displacement operations. Finally, we measure the strength of the displacement operations and observe the ratios (see Methods). In principle, we can also implement other ratios of displacement operations by applying additional $\sigma$- and $\pi$-polarized Raman beams (see Methods).

The second and third parts are realized by the red- and the blue-sideband transitions as shown in Fig. 2(e). The time-dependent strength-coefficient $g(t)$ in equation (3) is implemented by the change of laser intensity, which is proportional to the RF power on the AOMs of Fig. 2(a). We generate the time-dependent RF signal from an arbitrary waveform generator (AWG) and apply it to the AOM R2. By using the AWG, we can generate all the necessary RF frequencies and powers, which

realizes the full Hamiltonian (7) containing the displacement operation, red-, and blue-sideband transitions.

**Experimental procedure of the quantum simulation**. In the experiment, we initialize the motional and internal state of the ion to the state $\left|0_f 0_{\bar{f}}, n = 0\right\rangle$ by standard Doppler cooling, resolved sideband cooling, and optical pumping[34, 35]. The residual average phonon number and the heating rate are measured to be $\langle n \rangle = 0.016 \pm 0.025$ and $3.8 \pm 1.2$ quanta s$^{-1}$, respectively. The heating effect can be neglected in the typical duration of a single simulation, which is of $< 2$ ms. Then we implement the target Hamiltonian (7) and let the system evolve for a time $t$. Finally, we measure the average boson number $\langle n \rangle$ and the populations of various fermionic states, as well as the correlation between the bosonic mode and the fermionic state. A detailed discussion of the measurement procedures can be found in the Methods section. We compare the experimental results with the ideal theoretical calculations. In our simple situation of single bosonic, fermion, and anti-fermion modes, we are able to numerically calculate the exact evolution with the full Hamiltonian and find a perturbation method that works for a short time dynamics. The whole evolution is then computed by accumulation of the latter. We note that such numerical methods would not be allowed as the system size grows. Typically, one considers the size corresponding to 50 qubits to be intractable. For example, a realistic situation with 16 ions, 16 modes, and 8 considered levels per mode would be beyond the capabilities of classical computers.

**Self-interaction and particle creation and annihilation**. We first study the fermion self-interaction processes by setting $g_2 = 0$, starting from the initial state $\left|1_f 0_{\bar{f}}, 0_b\right\rangle$. Then the self-interacting dynamics occurs via the couplings $\left|1_f, 0_{\bar{f}}, n_b\right\rangle \to \left|1_f, 0_{\bar{f}}, n_b \pm 1\right\rangle$. Figure 3(a) shows experimental data for the time-dependent bosonic vacuum populations and the average boson numbers for different self-interaction strengths $g_1/\omega_0 = 0.1$ and $0.15$, which quantitatively coincide with the theoretical calculations within experimental errors. We clearly observe the expected emission and reabsorption processes of virtual bosons and the growth of the average number of virtual bosons with the self-interaction strength $g_1$.

Subsequently, we realize the annihilation of a fermion–antifermion pair and the creation of bosons with parameters $g_1 = 0.01\omega_0$, $g_2 = 0.21\omega_0$, and $\sigma_t = 3/\omega_0$. We choose the initial state to be the state with a fermion–antifermion pair and no bosons: $\left|1_f 1_{\bar{f}}, n = 0\right\rangle$. Figure 3(b) shows the dynamics of the fermion–antifermion scattering process via the population of the fermionic-pair state and the average boson number. It can be clearly seen that the initial fermion–antifermion pair disappears, creating a single boson.

Next, we realize the process of scattering with parameters $g_1 = 0.1\omega_0$, $g_2 = \omega_0$, $\sigma_t = 4/\omega_0$, where $g_2 \geq \omega_0$. In this regime, the interaction Hamiltonian ((2)) cannot be regarded as a perturbation. In such a strong coupling situation, we cannot easily discriminate the contributions from the self-interaction and pair production processes. When the initial fermion–antifermion pair disappears, more than a single boson is created in the process as shown in Fig. 3(c), which is qualitatively different from the dynamics shown in Fig. 3(b). As the size of the Hilbert space is not too large, we numerically calculate the dynamics of the Hamiltonian by direct numerical integration (see Methods), which is in agreement with the experimental results shown in Fig. 3(c). However, as the number of fermion–antifermion pairs and bosons increases, the exact numerical calculation will be intractable by classical means. We have also developed a perturbation method based on the observation that, for a

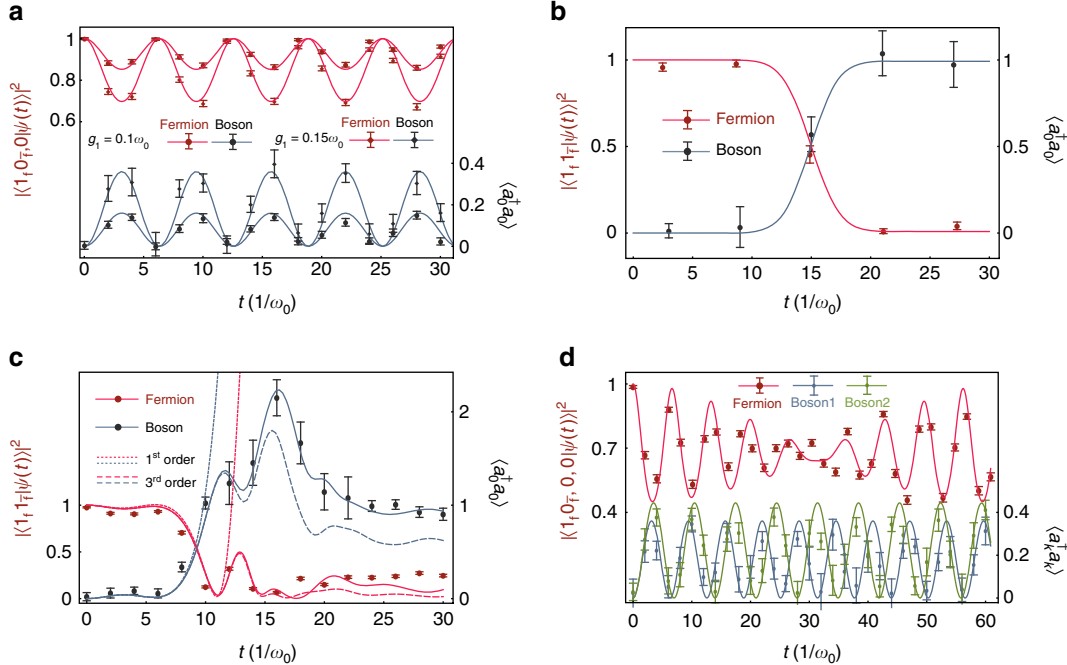

**Fig. 3** Trapped-ion simulation results of QFTs. Dots are experimental data and lines are numerical simulation curves. **a** Self-interaction process for parameters $g_1 = 0.1\omega_0$, $0.15\omega_0$, $g_2 = 0$, $\delta = 0$, and $T = 30/\omega_0$, where $|\psi(t)\rangle$ is the state at time $t$, evolved from $|\psi(0)\rangle = |1_f 0_{\bar{f}}, n = 0\rangle$. Red curves and left axis are for the population of state $|1_f 0_{\bar{f}}, n = 0\rangle$. Blue curves and right axis are for the average number of virtual bosons $\langle a_0^\dagger a_0 \rangle$. **b** Fermion and antifermion annihilation process for parameters $g_1 = 0.01\omega_0$, $g_2 = 0.21\omega_0$, $\sigma_t = 3/\omega_0$, where $|\psi(t)\rangle$ is the state evolved from $|\psi(0)\rangle = |1_f 1_{\bar{f}}, n = 0\rangle$. Red curves and left axis are for the population of state $|1_f 1_{\bar{f}}\rangle$. Blue curves and right axis are for the average number of bosons $\langle a_0^\dagger a_0 \rangle$. **c** The process in the strong coupling, where both of self-interaction and pair production processes strongly influence on the dynamics. Parameters are $g_1 = 0.1\omega_0$, $g_2 = \omega_0$, and $\sigma_t = 4/\omega_0$, and the initial state the same as for **b**. Solid lines are obtained by exact numerical simulation using the built-in solver of the ordinary differential equation in Mathematica (see Methods). Dashed lines are computed by a Dyson series expansion with Feynman diagrams up to 1st and 3rd orders after dividing the whole time by 100 (see Methods). By including the Dyson series up to the 7th order, the deviations from the exact numerical calculation below $10^{-4}$ (see Methods and Fig. 4). **d** Self-interaction process for two bosonic modes $\omega_1 = \omega_0$, $\omega_2 = 0.9\omega_0$, with parameters $g_1 = 0.15\omega_0$, $g_2 = 0$, $\delta = 0$, and $T = 30/\omega_0$, where $|\psi(t)\rangle$ is the evolved state from $|\psi(0)\rangle = |1_f 0_{\bar{f}}, n_1 = 0, n_2 = 0\rangle$. Red curves and left axis are for population of $|1_f 0_{\bar{f}}, n_1 = 0, n_2 = 0\rangle$. Blue, green curves, and right axis are for average number of virtual bosons $\langle a_k^\dagger a_k \rangle$, $k = 1$, 2. All error bars in experimental data above represent the SD of 100 measurements

reasonably small time $g_2 t \ll 1$, the effect of the coupling term $g_2$ does not produce a divergence in the dynamics. We divide the total time of the process by 100 and apply the perturbation method (see Methods) to the unitary evolution operator in each time slice. We find that after including terms up to the 7th order in the perturbation parameter, the deviation of the perturbative dynamics from the complete one is below $10^{-4}$. However, even this approach, based on a perturbative expansion within time slices, would be difficult to use for large Hilbert space dimensions with more fermions and bosons.

Finally, as a demonstration of scalability, we realize fermion self-interaction processes extended to two bosonic modes by using both $X$ and $Y$ phonon modes of a single trapped ion. We set $g_1 = 0.15\omega_0$, the first boson mode frequency $\omega_1 = \omega_0$, and the second boson mode frequency $\omega_2 = 0.9\omega_0$. We note that the $g_1$ ($g_2$) coupling to the mode $Y$ ($X$) is negligible, as the detuning to the mode $Y$ ($X$) is larger by a factor of 50, which effectively suppresses the strength by the same amount. We choose the initial state to contain one fermion and no bosons, $|1_f 0_{\bar{f}}, n_1 = 0, n_2 = 0\rangle$. Then the self-interacting dynamics is given by the transition $|1_f 0_{\bar{f}}, n_1, n_2\rangle \leftrightarrow |1_f 0_{\bar{f}}, n_1 \pm 1, n_2 \pm 1\rangle$. As the bosonic modes have different frequencies, we observe that the considered fermion emits and reabsorbs bosons differently from the single-boson case. Instead of a sine curve, we see a clear beat-note behavior of the fermionic population as shown in Fig. 3(d). We also clearly observe the dynamics of both bosonic modes in a good agreement with the theoretical expectation. By increasing

the number of bosonic modes, we would simulate the continuous regime of bosonic modes, which would be related to scattering experiments. In such large number of bosonic modes, the non-perturbative behavior of fermionic or antifermonic mode could be intractable. On the way of increasing bosonic modes, a technology of correlation measurement of multiple phonon modes could be applied[36, 37].

## Discussion

In conclusion, this work considers an experimental quantum simulation of interacting fermionic and bosonic quantum field modes. Our approach could be in principle scaled up by progressively incorporating more fermionic and bosonic field modes, which may lead to a full-fledged digital-analog quantum simulation of QFTs such as QED[22–24] or the Holstein model[38], where correlations between multiple fermions and phonons have critical relevance. In our current experimental system, an extension to multi-fermion and multi-phonon (bosonic) modes could in principle be implemented by loading a number of ions in a single trap, where the spins of ions map the fermionic modes through Jordan–Wigner transformation[24] and the vibrational modes of ions directly map the bosonic modes. The many-body operators or spin–spin interactions appearing after mapping of the fermionic modes onto spins can be efficiently implemented via a combination of two Mølmer–Sørensen gates and a local gate as shown in ref.[24]. Other than the spin–spin interactions in the Holstein model, e.g., the couplings between fermionic modes and

bosonic modes can be implemented by the same Raman laser beams that are individually addressing single ions and tuned to specific mode frequencies. In this respect, it has been shown that the number of gates grows polynomially as the number of fermions and bosons[38]. Ref.[38] also discussed the estimated infidelities from the gate errors in realistic experimental decoherence condition up to four sites, which clearly showed the degree of control is more demanding when the coupling strengths between the modes increase. As demonstrated in our experiment, we do not observe any clear degradation of the simulation when using two modes, although here we do not have the technical problem of individual addressing. We may implement the model in a fully analogue way together with proper spin–spin interactions[39–41], which would allow us to study the pairing or polaron physics occurring in many unconventional superconducting systems[42, 43] with the controls of various parameters. In particular, we remark that already with 16 two-level ions and 8 phononic levels per ion, one could perform quantum simulations of interacting quantum field modes that are beyond the reach of classical computations, i.e., a Hilbert space dimension of $16^{16} \sim 2^{64}$, which would otherwise require a lengthy quantum algorithm with 64 qubits[44, 45]. This experiment opens an avenue that aims at out-performing the limitations of classical computers, with in principle scalable quantum simulations.

We also point out that there are no known efficient classical algorithms for simulating interacting fermionic models in arbitrary spatial dimensions, whereas with our approach, with a trapped-ion quantum simulator, fermionic models in arbitrary dimensions could be analyzed with polynomial resources[24]. The verification of our proposed scalable experiment requires polynomial resources, as for the detection of the number of bosonic excitations produced, or the population of the fermionic or antifermionic states, only a polynomial number of measurements is required.

## Methods

**Uniform red sideband.** The "uniform red sideband"[46–48] is implemented as an adiabatic transition where the transfer speed between $\left|0_f\bar{0}_{\bar{f}}, n\right\rangle$ and $\left|1_f\bar{1}_{\bar{f}}, n-1\right\rangle$ is the same for all $n = 1, 2, \ldots$. It is realized by adding a time-dependent amplitude $A(t) = \sin(\pi t/d)$ and a time-dependent phase $\varphi(t) = -1/\beta \sin(\pi t/d)$ to the normal red-sideband operation, and some additional terms to compensate for the AC Stark shift. Here, $d = c\,\pi_{\text{red}}$ is the duration of the transition, and $\beta = ((l+1)(h+1))^{-1/4}/c$ is an empiric parameter depending on the lower bound $l$ and the upper bound $h$ of the phonon number $n$. We typically choose $c$ to be 10, such that the transition duration $d$ is $c/2 = 5$ times the red-sideband operation period. Therefore, we achieve more than 99% of theoretical fidelity for all phonon numbers between $l$ and $h$.

**Displacement strength adjustment.** We experimentally measure several strength coefficients to check the strength ratios depending on the electronic states. We first prepare the initial state $|m, n = 0\rangle$, where $|m\rangle$ is either $\left|0_f\bar{0}_{\bar{f}}\right\rangle$ or $\left|1_f\bar{0}_{\bar{f}}\right\rangle$. Then we apply the displacement operation for a small period $\tau$. After this, we should obtain a coherent state $|m, \alpha\rangle$, $\alpha = \Omega\tau$, where $\Omega$ is the desired strength coefficient. Subsequently, with several different $\tau$, we fit the parameter $\Omega$ by measuring each time the remaining population on state $|m, n = 0\rangle$ with the "uniform red sideband" method, which should be $e^{-\alpha^2} = e^{-\Omega^2\tau^2}$. After careful beam alignment and quarter wave plate adjustment, the measured strength coefficients of $\left|0_f\bar{0}_{\bar{f}}\right\rangle$ and $\left|1_f\bar{0}_{\bar{f}}\right\rangle$ are $(2\pi)7.2$ and $(2\pi)14.4$ kHz, respectively, which are consistent with the theory ratio 1:2.

As the magnetic quantum number is conserved during the displacement operation, two virtual optical transitions in a Raman path should have the same polarization. If both polarizations are purely $\sigma_-$, then the relative strength-coefficient ratio between states $\left|0_f\bar{0}_{\bar{f}}\right\rangle$, $\left|1_f\bar{0}_{\bar{f}}\right\rangle$, $\left|1_f\bar{1}_{\bar{f}}\right\rangle$, and $\left|0_f\bar{1}_{\bar{f}}\right\rangle$ is $1:0:1:2$. If both polarizations are purely $\pi$, then the relative strength-coefficient ratio between states $\left|0_f\bar{0}_{\bar{f}}\right\rangle$, $\left|1_f\bar{0}_{\bar{f}}\right\rangle$, $\left|1_f\bar{1}_{\bar{f}}\right\rangle$, and $\left|0_f\bar{1}_{\bar{f}}\right\rangle$ is $1:1:1:1$. In general, if the ratio between $\sigma_+$, $\sigma_-$, and $\pi$ polarization is $a:b:c$, then the relative strength-coefficient ratio between states $\left|0_f\bar{0}_{\bar{f}}\right\rangle$, $\left|1_f\bar{0}_{\bar{f}}\right\rangle$, $\left|1_f\bar{1}_{\bar{f}}\right\rangle$, and $\left|0_f\bar{1}_{\bar{f}}\right\rangle$ is $a+b+c : 2a+c : a+b+c : 2b+c$.

**Fermionic state measurement.** To measure $P\left(\left|1_f\bar{1}_{\bar{f}}\right\rangle\right)$, we simply apply a $\pi$ rotation between states $\left|0_f\bar{0}_{\bar{f}}\right\rangle$ and $\left|1_f\bar{1}_{\bar{f}}\right\rangle$ to swap their populations, with a microwave horn. Then the measured population of state $\left|0_f\bar{0}_{\bar{f}}\right\rangle$ is equal to the original $P\left(\left|1_f\bar{1}_{\bar{f}}\right\rangle\right)$.

To measure $P\left(\left|1_f\bar{0}_{\bar{f}}, n = 0\right\rangle\right)$, however, we need a phonon projective measurement[46–48]. Instead of using fluorescence detection together with a post-selection scheme, which may introduce significant heating errors because of photon scattering, here we use an auxiliary state as a swap buffer. It is noteworthy that the interaction Hamiltonian (7) does not have terms related to state $\left|0_f\bar{1}_{\bar{f}}\right\rangle$. Therefore, we employ state $\left|0_f\bar{1}_{\bar{f}}\right\rangle$ as the auxiliary state and always initialize it to zero. We first apply three consecutive $\pi$ swap gates between $\left|0_f\bar{0}_{\bar{f}}\right\rangle$ and $\left|0_f\bar{1}_{\bar{f}}\right\rangle$, and between $\left|1_f\bar{1}_{\bar{f}}\right\rangle$ and $\left|1_f\bar{0}_{\bar{f}}\right\rangle$. After these operations, $\left|1_f\bar{1}_{\bar{f}}\right\rangle$ is swapped with $\left|1_f\bar{0}_{\bar{f}}\right\rangle$, and $\left|0_f\bar{1}_{\bar{f}}\right\rangle$ is swapped with $\left|0_f\bar{0}_{\bar{f}}\right\rangle$. Then we apply a "uniform red sideband" $\pi$ rotation to swap the population in $\left|0_f\bar{0}_{\bar{f}}, n>0\right\rangle$ with that of $\left|1_f\bar{1}_{\bar{f}}, n-1\right\rangle$. Then, we measure the remaining vacuum-state population, $P\left(\left|0_f\bar{0}_{\bar{f}}, n = 0\right\rangle\right)$, which is equal to the original population, $P\left(\left|1_f\bar{0}_{\bar{f}}, n = 0\right\rangle\right)$. The uncertainty of the measurement mainly comes from the quantum projection noise of binary result of single measurements[49].

For the experiment involving two boson modes, we first measure $P\left(\left|1_f\bar{0}_{\bar{f}}, n_1 = 0\right\rangle\right)$ using the same method as that of the single-boson case. Next we consecutively apply a "uniform red sideband" to the first mode and another "uniform red sideband" to the second mode. Then, we measure the population of the upper state, which should be $P\left(\left|1_f\bar{0}_{\bar{f}}, n_1 = 0, n_2 > 0\right\rangle\right)$. Therefore, we obtain the desired population from the relation: $P\left(\left|1_f\bar{0}_{\bar{f}}, n_1 = 0, n_2 = 0\right\rangle\right) = P\left(\left|1_f\bar{0}_{\bar{f}}, n_1 = 0\right\rangle\right) - P\left(\left|1_f\bar{0}_{\bar{f}}, n_1 = 0, n_2 > 0\right\rangle\right)$. It is noteworthy that this scheme is clearly scalable in the number of bosonic modes.

**Average boson number measurement.** For the average boson number measurement, we first use optical pumping to trace out electronic states[48] and then apply a blue sideband time sweep from $t = 0$ to $t = 12$ $\pi_{\text{blue}}$[50]. We get the phonon number distribution by fitting the result signals through the maximum likelihood method with parameters of the Fock state populations[46, 47]. The main uncertainty in the average phonon measurement comes from fitting and we include one standard deviation as an uncertainty throughout the manuscript.

**Ideal theoretical calculations.** The exact dynamics of the Hamiltonian (7) can be obtained by solving the time-dependent Schrödinger equation $i\hbar\frac{\partial}{\partial t}|\psi(t)\rangle = H_1(t)|\psi(t)\rangle$. We numerically solve the equation with the built-in function of Mathematica, NDSolve, which finds a numerical solution to the ordinary differential equation mainly based on Runge–Kutta method. In the numerical calculation, we include 4 internal levels and up to 10 phonons per mode for the Hamiltonian (7), which changes the Schrödinger equation to the ordinary differential equation with 40 and 400 components for single mode and two modes of the state $|\psi\rangle$, respectively. With the option of infinite Maxsteps in Mathematica, the numerical calculations converge and do not show any error messages. We point out that all the parameters in the simulation are experimentally determined, not obtained via fitting. The main limitation of the numerical calculation would be the size of the Hilbert space when we scale up the system with multiple fermions and bosons.

**Feynman diagram calculation.** In the interaction picture, the evolution operator $U_1(t, t_0)$ satisfies the following differential equation

$$i\hbar\frac{\partial}{\partial t}U_1(t, t_0) = H_1(t)U_1(t, t_0), \tag{11}$$

which can be exactly solved as the so-called Dyson series,

$$U_1(t, t_0) = \sum_{n=0}^{\infty}\left(-\frac{i}{\hbar}\right)^n\int_{t_0}^{t}dt_1\ldots\int_{t_0}^{t_{n-1}}dt_n H_1(t_1)\ldots H_1(t_n). \tag{12}$$

By introducing the time-ordering operator $\mathcal{T}$, the above solution can be written in a formally succinct way $U_1(t, t_0) = \mathcal{T}\exp\left(-\frac{i}{\hbar}\int_{t_0}^{t}H_1(s)ds\right)$.

Truncating at some finite $N$ in equation (12) provides a straightforward perturbation treatment of the evolution operator $U_1(t, t_0)$. However, when the evolution time increases, the unitarity of the perturbation expansion becomes difficult to guarantee, because the truncation error is proportional to $(t - t_0)^{N+1}$. In order to deal with the long-time dynamics, we make use of the composition property of the evolution operator and interleave $M - 1$ equally spaced points between $t_0$ and $t$. Then, the evolution operator $U_1(t, t_0)$ is identically written as the product of $M$ evolution operators, each of which governs the dynamical evolution over a short period of time,

$$U_1(t_0, t) = \prod_{m=1}^{M}U_1(t_m, t_{m-1}), \tag{13}$$

with $t_M \equiv t$. For any dynamics with finite duration, saying $t - t_0$ is finite, we can always assign a sufficiently large $M$ so that $\Delta t \equiv t_m - t_{m-1}$ is a small but finite quantity. Consequently, $U_1(t_m, t_{m-1})$ is readily to be treated perturbatively.

**a**

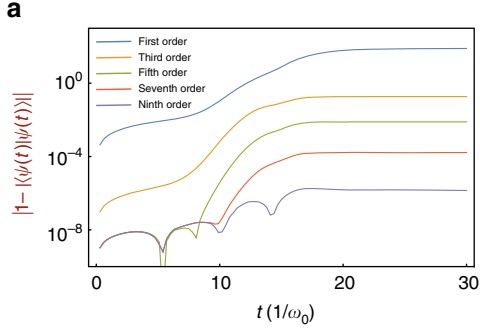

**b**

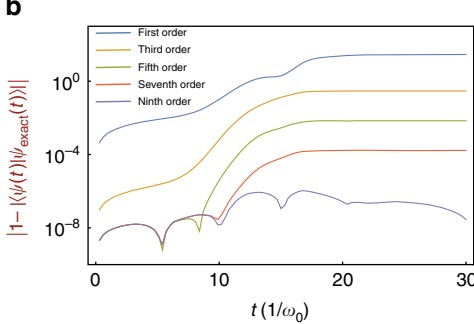

**Fig. 4** Convergence of Dyson series. In order to evaluate the validity of the perturbation calculations, we use (**a**) the norm of the state from 1, $|1 - |\langle \psi(t)|\psi(t)\rangle||$ and (**b**) the infidelity of the state, $|1 - |\langle \psi_{exact}(t)|\psi(t)\rangle||$, where $\psi_{exact}(t)$ is the result of the ideal numerical calculation, for the case shown in Fig. 3(c) with $M = 100$ divisions of time. After including up to 7th order perturbation, the deviation of the norm from 1 and the infidelity reduce to below $10^{-4}$. We note that for the case of Fig. 3(b), even the first and the second order perturbations provide the deviation of the norm and infidelity below $5 \times 10^{-2}$ and $2 \times 10^{-4}$, respectively

Denote the $n$-th order perturbation expansion of $U_I(t_m, t_{m-1})$ as $U_I^{(N)}(t_m, t_{m-1})$,

$$U_I^{(N)}(t_m, t_{m-1}) = \sum_{n=0}^{N} \left(-\frac{i}{\hbar}\right)^n \int_{t_{m-1}}^{t_m} ds_1 \dots \int_{t_{m-1}}^{s_{n-1}} ds_n H_I(s_1) \dots H_I(s_n). \quad (14)$$

Then the whole dynamics can be treated perturbatively as follows,

$$U_I(t, t_0) = \prod_{m=1}^{M} U_I^{(N)}(t_m, t_{m-1}) + \mathcal{O}\left(\frac{(t - t_0)^{N+1}}{M^N}\right). \quad (15)$$

The deviations are related to the number of sliced sections $M$ in time and the order of perturbations $N$. In our numerical calculations of the Dyson series, we divide the total time by $M = 100$ and apply the perturbations up to $N = 9$th order. Figure 4 shows the deviations of the norm from 1 $\left|1 - |U_I(t, t_0)|\psi(t_0)\rangle|^2\right|$ and the infidelity $|1 - |\langle \psi_{exact}(t)|U_I(t, t_0)|\psi(t_0)\rangle||$ depending on the order of the perturbations for the case of Fig. 3(c). Here $\psi_{exact}(t)$ is the result by the ideal numerical calculation. From the 7th order, the deviations in the perturbation calculation are below $10^{-4}$ from the ideal norm of 1.

**Data availability**. The data that support the findings of this study are available from the corresponding author on request.

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

## Acknowledgements

This work was supported by the National Key Research and Development Program of China under Grants Number 2016YFA0301900 (Number 2016YFA0301901), the National Natural Science Foundation of China 11374178, 11405093, 11574002, and 11504197, as well as Spanish MINECO/FEDER FIS2015-69983-P, Ramón y Cajal Grant RYC-2012-11391, UPV/EHU UFI 11/55, Project EHUA14/04, Basque Government grant IT986-16, a UPV/EHU PhD fellowship, the Alexander von Humboldt foundation, EU STREP EQUAM, and ERC Synergy grant BioQ. M.-H.Y. acknowledges support from National Natural Science Foundation of China 11405093, the Guangdong Innovative and Entrepreneurial Research Team Program (Number 2016ZT06D348), and the Science Technology and Innovation Commission of Shenzhen Municipality (ZDSYS20170303165926217 and JCYJ20170412152620376). X.Z. acknowledges support from the National Postdoctoral Program for Innovative Talents, Number BX201601908.

## Author contribution

X.Z. and K.Z. with the support of Y.S. and S.Z. performed the experiments and the data taking. X.Z. and K.Z. analyzed the data. X. Z. and J.-N.Z. developed the method of time-sliced perturbation calculation. J.-N.Z., M.-H.Y., J.C, J.S.P., L.L., and E.S. provided the theoretical support. K.K. supervised the project. All authors contributed to the writing of the manuscript.

## Additional information

**Competing interests:** The authors declare no competing financial interests.

