## [Peer Review File · Nature Communications]

Reviewers' comments:

Reviewer #1 (Remarks to the Author):

The manuscript "Fermion-antifermion scattering via boson exchange in a trapped ion" describes an experimental implementation of a quantum simulation originally proposed in Ref 23. Experimentally the quantum simulation involves 4 internal energy levels of a Yb⁺ ion coupled to up to two motional degrees of freedom. This is a sufficiently small system that the experimental results can be calculated very accurately. In addition, the operations that are implemented on the trapped Yb⁺ ion are similar to what has been implemented in trapped ion experiments for many years. Nevertheless, the system that is being simulated (fermion-antifermion scattering via boson exchange) is novel, and I believe the manuscript, which is well written, provides a very nice description of how the different operations in the experiment can be interpreted in quantum field theory language. As such I believe this manuscript will be of significant interest to physicists working with trapped ions as well as the quantum simulation community in general, and will motivate further work simulating quantum field theories with trapped ions. In summary, I believe the manuscript is suitable for publication in Nature Communications, if the authors can provide satisfactory replies to the following comments.

1. The quantum simulation includes two interactions, a self-interaction term and a pair creation term. Equation (6) lacks a self-interaction term for the antifermion state $|0_f, 1_{\bar{f}}\rangle$. Why is this? Physically shouldn't all fermion and anti-fermion states have a self-interaction?
2. The authors claim a demonstration of scalability by extending the simulation from one mode (i.e boson) to two modes (bosons). I find this a rather soft demonstration. In particular, scalability requires demonstrating how the number of fermions can be increased as well. Presumably this requires working with more than one ion and with 4 energy levels in each ion. Simulating multiple fermions then requires the fermions to have an overall anti-symmetric wave function. Some discussion of how this will and can be done should be given in the manuscript. This directly impacts the potential interest in this manuscript by the quantum simulation community.
3. In the methods I was not previously aware of the uniform sideband method which is briefly described. Is this the first implementation of this method? If not, a reference is appropriate.
4. In the methods section, the section on the Feynman diagram calculation discusses a need to rescale the interaction Hamiltonian. I was not aware of this protocol. A reference could be helpful.

Reviewer #2 (Remarks to the Author):

The manuscript of X. Zhang et al, entitled “Fermion-antifermion scattering via boson exchange in a trapped ion” describes the experimental realization of a proposal by J. Casanova et al. [Phys. Rev. Lett. 107, 260501 (2011)]. The authors claim the demonstration of fermion-antifermion scattering simulation in a simplified model using a single trapped atomic ion. In turn, the simulated model incorporates elementary features of quantum electrodynamics. In summary their work is described as a proof of principle experiment on simulating quantum field theories, in a novel, efficient, and scalable way.

Overall, their work is judged to be valid and claims are significant to their field of research. Their implementation of interactions on multiple internal and external degrees of freedom of trapped ions is certainly very interesting for the community of experimental quantum simulations. This work, together with the proposal [Phys. Rev. Lett. 107, 260501 (2011)], will likely trigger further attention on simulating features of quantum field theories and drive developments toward new approaches and methods both in theory and experiments. However, the authors do not convincingly discuss the prospects of future, more complex experimental simulations that are intractable by numerical simulations, which is a common problem in claiming scalable techniques.

Certainly, the manuscript would improve when it is more carefully and clearly written; detailed comments follow below. In particular, discussing details of systematic effects and corresponding limitations of their approach would enhance the credibility of their claim of scalability. Further, it would be important to discuss more details on the experiments and the performed calculations in the Methods section. In addition, more references to original work, which is employed for their methodology, is needed. Consequently, after consideration of these issues and improvement of transparency the author's work can be reasonably judged even by non-specialists and the manuscript can be more sufficiently used to drive developments in other groups.

To conclude, a revised version of the manuscript will be suitable for publication in Nat. Commun.

In case of a revision of the manuscript, the authors may also consider the following more specific suggestions:

- Authors should re-edit Figures to ensure proper display. For example, in Fig.1a avoid mixing mathematical symbols with drawings, and Fig. 2 is not properly formatted and some labels display too small to be visible. Moreover, Fig. 3 is to show solid and dashed lines of different colors to indicate numerical simulations (solid lines) and Dyson-series expansion calculations (dashed lines), however, agreement between both in Fig. 3a,b, and d is not visible; Legend labels and figure caption need to be improved.

- The theoretical considerations presented on pages 3 and 4 seem to be very close to the cited Ref. [23] and long parts read like repetitions of paragraphs of the original work. It may be possible to be more concise on the theory part and instead elaborate on important differences in the authors implementation and the original proposal.

- On Page 5, "The operations of the self-interaction and scattering processes [...] are realized by [...] Raman laser beams [...]", Raman laser beams is slang and needs Reference.

- On Page 5, "The train of laser pulses in the time domain can be considered as an equally spaced frequency "comb" [...]", needs Reference.

- An example of less careful writing style: On Page 6, "The remaining steps are the standard sideband operations with [...]", "[...] implemented by modulating the amplitude of the laser beams [...]", and "By using the AWG, we are also able to add up signals by mathematical calculations at the sampling stage.". What do the authors mean by "standard" and "amplitude of laser beams", and what is the last sentence suppose to describe?

- On Page 6, "In the experiment, we initialize the motional and internal state [...] by standard Doppler cooling, resolved sideband cooling and optical pumping." needs a Reference.

- An example, where experimental results are quoted without giving uncertainties: On Page 6, "The residual average phonon number and the heating rate are measured to be $\langle n \rangle = 0.016$ and 3.8 quanta s^{-1} , respectively." Authors should indicate uncertainties, in particular, for non-specialists, who may not be able to judge significances of results for themselves.

- On page 7, "Fig. 3(a) shows experimental data [...] quantitatively coincide with the theoretical calculations within experimental errors." In general, in the Results part of the manuscript, it should be concisely stated what kind of "theoretical calculations" have been performed, while in the Methods sections their details and limitations should be discussed. The "exact quantum dynamics",

on page 7, is not described in the methods section, while the description of the “20th-order perturbative calculations” is not sufficiently discussed. Why did the authors choose the 20th order? Can they visualize that perturbation calculations will not converge in the presented case (Fig. 3c) in comparison to cases where it does (e.g., Fig. 3b)? What kind of numerical calculation are performed to yield “exact quantum dynamics” and what are corresponding uncertainties and limitations for this?

- On page 7, “Finally, as a demonstration of scalability, we realize fermion self-interaction processes extended to 2 bosonic modes by using both X and Y phonon modes of a single trapped ion.” While this is of course a first step in scaling the system, it should be carefully discussed what systematic effects come into play when increasing the number of participating modes. What kind of limitations are expected to emerge when increasing the number of modes even more? For example, how do the authors propose to significantly extract Fock state population distributions for an increasing number of modes? Why are experimental results for average number of virtual bosons in Fig. 3d missing?

- On page 8, “ In particular, we remark that already with 10 two-level ions and 5 phononic levels per ion, one could perform quantum simulations of interacting quantum field modes that are beyond the reach of classical computations, that is, a Hilbert space dimension of $10^{10} \sim 2^{33}$, which would otherwise require a lengthy quantum algorithm with 33 qubits.” This statement is very strict and leaves no room for developments in efficient (classical) numerical approaches based on effective theories or models for the dynamics. Further, the authors should note that any complex simulation result has a verification problem as stated in criterion 5, discussed in Ref. [Cirac, J. I. & Zoller, P. Goals and opportunities in quantum simulation. Nat. Phys 8, 264–266 (2012)]. How would the authors suggest on dealing with this issue in their approach?

- In the Methods section, relevant details on the experimental techniques are not cited properly and uncertainties are not discussed, in particular, in the section about “Fermionic state measurement” and “Average boson number measurement”. Authors should cite original work, which introduced used methods, and they should discuss statistical as well as systematic uncertainties in the their experiment. In particular, it is not valid to just write the following: “After careful beam alignment and quarter wave plate adjustment, the measured strength coefficients of [...] are $(2\pi)7.2$ kHz and $(2\pi)14.4$ kHz, respectively, which are consistent with the theory ratio 1: 2.” How large are the uncertainties on the experimental rates and how would a deviation from that 'theory ratio' show up in their results?

Reviewer #3 (Remarks to the Author):

In their manuscript, Zhang and coworkers report on a trapped ion experiment, where a generalized Jaynes-Cummings model is realized using four internal states of a trapped Yt ion, coupled to vibrational modes. The experiment follows closely previous proposals, in particular Ref. [23], where the microscopic Hamiltonian is mapped onto a minimal field theory made of one particle with fermionic statistics, one anti-particle, and bosonic fields. The main difference with respect to the initial proposal seems to be that here a single ion is used, instead of two, mapping the four available fermionic states onto 4 internal states. After an initial state preparation of the system, the authors let the system evolve (coherently, given the relatively slow dissipation rate) and perform measurements of the state populations of both fermions and bosons, and some basic correlation functions. They observe significant deviations from both the experiment and exact numerics with respect to perturbation theory - the latter cannot capture the system dynamics after long times.

From the experimental viewpoint, the work is sound. Several details are provided, and the prescription to load the initial state and perform the time evolution are well explained. From the theoretical viewpoint, the mapping from Eq. (6) to the ion and bosonic modes is clearly explained. What is less clear is the mapping from Eq. (1) to Eq. (2-3) - in particular, one wonders why a time-independent Hamiltonian operator is mapped onto a time-dependent one. This seems to be covered in Ref. [23]; nevertheless, a qualitative explanation on how this actually works would be highly desirable even in the present manuscript.

While I believe the work reported is valid, in my opinion, its results do not match the novelty and impact criteria of Nature Communications. Let me detail below my criticisms.

1) The experiments simulate the physics of a rather simple model Hamiltonian. In this respect, this work constitutes a natural follow up of the remarkable simulation of the Dirac equation proposed and realized by some of the authors in Refs. [10,11].

The main novel aspect here is the additional coupling to bosonic modes. However, this is rather different from what naturally occurs in quantum field theories, where the bosonic degrees of freedom are gauge couplings - as reported, e.g., in Ref. [21], consisting of two pairs of particles and antiparticles.

More importantly, I think the statement made by the authors, that their work is the first experimental realization of a trapped ion system simulating the dynamics of Hamiltonians involving both fermions and bosons, is not correct - this was done, including gauge symmetries, already in Ref. [21]. This somehow severely mitigates the novelty and impact of the present work - there have been already building block experiments showing how trapped ions can indeed serve as quantum simulators of quantum field theories including both fermions and (gauge) bosons.

2) Given the previous consideration, one could wonder whether the present scheme can nevertheless be useful in understanding some not-easy-to-simulate physical phenomena. The authors provide a simple demonstration that this could potentially be the case, by showing that the system can cope well with strong coupling - a regime where controlled numerics are challenging. However, the main question one has is how well this system is scalable, and what questions can really be answered in the close future.

2a - Scalability of the approach is not discussed at all. In particular, a detailed feasibility study should be provided to demonstrate that larger systems can be used to faithfully reproduce the target dynamics. This point is critical, in particular given the fact that it would be hard to control at the same time a relatively large amount of couplings following the present scheme. This even without considering the fact that, at present, only one-dimensional systems can be accessed with reasonable tunability and fidelities.

2b - Another important point which is not discussed is whether the present approach is opening new doors to solve unsettled problems. In the conclusions, the authors state that considering 10-ions systems, they could tackle problems at the boundary of exact diagonalization. (one could argue that this is not beyond classical computations, as matrix-product state methods can handle relatively easily the time evolution of such small one-dimensional systems, for the case of field theories see, e.g., Banuls et al., POS (Lattice 2013) 332). However, it is unclear which open, relevant problem can actually be answered in this context - what is somebody ultimately learning by these experiments if they can indeed be scaled up to 10 ions?

In view of these last two criticisms, the impact of the present paper is very unclear, especially outside of the trapped ion community.

In summary, I believe this is valid work, but it does not meet Nature Communications high standards of impact and novelty. I have few additional comments that the authors might find useful addressing for future submissions:

a) the word 'antifermionic' is rather uncommon. While its meaning is clear within the present context, I'd suggest to change wording, and refer to the fermionic modes as particle and antiparticle.

b) for quantum simulators to approach field theories, one typically requires Lorentz-invariance to be recovered at least in some limits. How can one see this aspect in the present implementation?

c) in the pdf version that I got, there are no experimental points in Fig. 3d for both the green and blue curves, not sure why.

Reply to Reviewer #1:

We thank the Reviewer for his/her positive assessment of our paper and the suggestions for improvement, which we have fully implemented. Please find below, after each Reviewer query (in black), our corresponding reply and manuscript modifications (in blue)

Reviewers' comments:

Reviewer #1 (Remarks to the Author):

The manuscript “Fermion-antifermion scattering via boson exchange in a trapped ion” describes an experimental implementation of a quantum simulation originally proposed in Ref 23. Experimentally the quantum simulation involves 4 internal energy levels of a Yb+ ion coupled to up to two motional degrees of freedom. This is a sufficiently small system that the experimental results can be calculated very accurately. In addition, the operations that are implemented on the trapped Yb+ ion are similar to what has been implemented in trapped ion experiments for many years. Nevertheless, the system that is being simulated (fermion-antifermion scattering via boson exchange) is novel, and I believe the manuscript, which is well written, provides a very nice description of how the different operations in the experiment can be interpreted in quantum field theory language. As such I believe this manuscript will be of significant interest to physicists working with trapped ions as well as the quantum simulation community in general, and will motivate further work simulating quantum field theories with trapped ions. In summary, I believe the manuscript is suitable for publication in Nature Communications, if the authors can provide satisfactory replies to the following comments.

We thank the Reviewer for his/her positive comments.

1. The quantum simulation includes two interactions, a self-interaction term and a pair creation term. Equation (6) lacks a self-interaction term for the antifermion state $|0_f, 1_{\bar{f}}\rangle$. Why is this? Physically shouldn't all fermion and anti-fermion states have a self-interaction?

The reason for this is that, at first order, the one-antifermion state is a dark state of the Hamiltonian. This is due to the fact of the asymmetric role of the fermionic and antifermionic operators in the fermionic field: the fermionic operator appears as "b", while the antifermionic operator appears as "d[†]". Therefore, in the interaction Hamiltonian, the b[†] operator appears to the left of the b, while the d[†] operator appears to the right of the d. The consequence of this is that the one-antifermion state does not contribute to this dynamics at lowest order. (of course, when one considers higher orders in the Dyson expansion, and more fermionic and bosonic modes, a self-energy contribution for the antifermions will also appear.) We agree in that perhaps this was not totally clear in the previous version, so we have added the sentence, below Eq. (6):

"We point out that, due to the asymmetric role of fermionic annihilation and antifermionic creation operators in the fermionic field, the one-antifermion state is a dark state of the Hamiltonian in Eq. (2) and therefore the antifermion does not have self-energy at first order (it has, when considering more modes and higher orders)."

2. The authors claim a demonstration of scalability by extending the simulation from one mode (i.e boson) to two modes (bosons). I find this a rather soft demonstration. In particular, scalability requires demonstrating how the number of fermions can be increased as well. Presumably this requires working with more than one ion and with 4 energy levels in each ion. Simulating multiple fermions then requires the fermions to have an overall anti-symmetric wave function. Some discussion of how this will and can be done should be given in the manuscript. This directly impacts the potential interest in this manuscript by the quantum simulation community.

We agree that proving this is a crucial aspect of our formalism to quantum-simulate fermionic and bosonic modes in interaction and in arbitrary spatial dimensions. We have proved this scalability of our techniques for simulating fermionic and bosonic modes, for 2 and 3 spatial dimensions, in the articles

[24] J. Casanova, A. Mezzacapo, L. Lamata, and E. Solano, Quantum Simulation of Interacting Fermion Lattice Models in Trapped Ions, Phys. Rev. Lett. 108, 190502 (2012), and

[25] A. Mezzacapo, J. Casanova, L. Lamata, and E. Solano, Digital Quantum Simulation of the Holstein Model in Trapped Ions, Phys. Rev. Lett. 109, 200501 (2012).

To clarify this further, we included the paragraph, at the end of page 2/beginning of page 3:

"The scalability of our protocols for simulating fermionic models in two and three spatial dimensions was demonstrated in Ref. [24]. There, it is shown that the many-body operators appearing after mapping of the fermionic modes onto spins via Jordan-Wigner transformation, can be efficiently implemented via a combination of two M₀mer-S₀rensen gates and a local gate. We also

proved in Ref. [25] that one may add bosonic modes to the fermionic simulation in a scalable and efficient way, by means of a digital-analog quantum simulator.", where the new Ref. [25] is the paper by Mezzacapo et al, Phys. Rev. Lett. 109, 200501 (2012), and all previous references are therefore shifted one position.

3. In the methods I was not previously aware of the uniform sideband method which is briefly described. Is this the first implementation of this method? If not, a reference is appropriate.

We add the following 3 references:

[39] An, S., Zhang, J.-N., Um, M., Lv, D., Lu, Y., Zhang, J., Yin, Z.-Q., Quan, H. T., and Kim, K., Experimental test of the quantum Jarzynski equality with a trapped-ion system, Nature Phys. 11, 193 (2015).

[40] Um, M., Zhang, J., Lv, D., Lu, Y., An, S., Zhang, J.-N., Nha, H., Kim, M., and Kim, K. Phonon arithmetic in a trapped ion system. Nat. Commun. 7, 11410 (2016).

[41] Lv, D., An, S., Um, M., Zhang, J., Zhang, J.-N., Kim, M. S., and Kim, K. Reconstruction of the Jaynes-Cummings field state of ionic motion in a harmonic trap. Phys. Rev. A 95, 043813 (2017).

4. In the methods section, the section on the Feynman diagram calculation discusses a need to rescale the interaction Hamiltonian. I was not aware of this protocol. A reference could be helpful.

The main idea of the rescaling method is described in the method section. We are not aware of similar methods reported before us.

Reviewer #2 (Remarks to the Author):

The manuscript of X. Zhang et al, entitled "Fermion-antifermion scattering via boson exchange in a trapped ion" describes the experimental realization of a proposal by J. Casanova et al. [Phys. Rev. Lett. 107, 260501 (2011)]. The authors claim the demonstration of fermion-antifermion scattering simulation in a simplified model using a single trapped atomic ion. In turn, the simulated model incorporates elementary features of quantum electrodynamics. In summary their work is described as a proof of principle experiment on simulating quantum field theories, in a novel, efficient, and scalable way.

We agree with this statement by the Reviewer. Our paper is a novel, efficient, and scalable proposal for implementing interacting fermionic and bosonic modes towards a full-fledged realization of quantum field theories with trapped ions.

Overall, their work is judged to be valid and claims are significant to their field of research. Their implementation of interactions on multiple internal and external degrees of freedom of trapped ions is certainly very interesting for the

community of experimental quantum simulations. This work, together with the proposal [Phys. Rev. Lett. 107, 260501 (2011)], will likely trigger further attention on simulating features of quantum field theories and drive developments toward new approaches and methods both in theory and experiments. However, the authors do not convincingly discuss the prospects of future, more complex experimental simulations that are intractable by numerical simulations, which is a common problem in claiming scalable techniques.

We agree, as also mentioned to the Editor and Reviewer 1, in that our proposal is fully scalable, but perhaps this was not totally clear in the previous version. Therefore, we have added the paragraph at the end of page 2:

"The scalability of our protocols for simulating fermionic models in two and three spatial dimensions was demonstrated in Ref. [24]. There, it is shown that the many-body operators appearing after mapping of the fermionic modes onto spins via Jordan-Wigner transformation, can be efficiently implemented via a combination of two M₀lmer-S₀rensen gates and a local gate. We also proved in Ref. [25] that one may add bosonic modes to the fermionic simulation in a scalable and efficient way, by means of a digital-analog quantum simulator.", where the new Ref. [25] is the paper by Mezzacapo et al., Phys. Rev. Lett. 109, 200501 (2012), and all previous references are therefore shifted one number.

Certainly, the manuscript would improve when it is more carefully and clearly written; detailed comments follow below. In particular, discussing details of systematic effects and corresponding limitations of their approach would enhance the credibility of their claim of scalability. Further, it would be important to discuss more details on the experiments and the performed calculations in the Methods section. In addition, more references to original work, which is employed for their methodology, is needed. Consequently, after consideration of these issues and improvement of transparency the author's work can be reasonably judged even by non-specialists and the manuscript can be more sufficiently used to drive developments in other groups.

To conclude, a revised version of the manuscript will be suitable for publication in Nat. Commun.

We greatly appreciate your positive recommendation of our work. We followed your detailed comments and improved the presentation of the work with more carefulness and clarity.

In case of a revision of the manuscript, the authors may also consider the following more specific suggestions:

- Authors should re-edit Figures to ensure proper display. For example, in Fig.1a avoid mixing mathematical symbols with drawings, and Fig. 2 is not properly formatted and some labels display too small to be visible. Moreover, Fig. 3 is to show solid and dashed lines of different colors to indicate numerical simulations (solid lines) and Dyson-series expansion calculations (dashed lines), however,

agreement between both in Fig. 3a,b, and d is not visible; Legend labels and figure caption need to be improved.

We improve and re-edit the figures as suggested.

In Fig. 1a, we removed the mathematical symbols in the drawing. We also significantly improve the Fig. 2 and change the captions accordingly. For Figs. 3a,b, and d, we decide not to include the dashed lines by calculations of the Dyson-series and explain the agreement in the caption.

- The theoretical considerations presented on pages 3 and 4 seem to be very close to the cited Ref. [23] and long parts read like repetitions of paragraphs of the original work. It may be possible to be more concise on the theory part and instead elaborate on important differences in the authors implementation and the original proposal.

We believe the paper should be to a certain extent self-contained, in that it should contain enough theory formalism to make it understood without the need to read the theory paper in detail. Nevertheless, many details, as the comoving modes, and the complete derivation of the trapped-ion Hamiltonian, are omitted in the experimental paper. Therefore, we prefer to leave the theory part as it is now, to have a minimum of formalism in the experimental paper to make it better understood. We have also been more specific about details of the experiment to make clear the technicalities and specificities of this implementation.

- On Page 5, “The operations of the self-interaction and scattering processes [...] are realized by [...] Raman laser beams [...]”, Raman laser beams is slang and needs Reference.

We include the following references for the Raman laser beams:

[31] Monroe, C., Meekhof, D. M., King, B. E., Jefferts, S. R., Itano, W. M., Wineland, D. J., and Gould, P. Resolved-sideband raman cooling of a bound atom to the 3d zero-point energy. *Phys. Rev. Lett.* 75, 4011–4014 (1995).

[32] Roos, C., Zeiger, T., Rohde, H., Nagerl, H. C., Eschner, J., Leibfried, D., Schmidt-Kaler, F., and Blatt, R. Quantum state engineering on an optical transition and decoherence in a paul trap. *Phys. Rev. Lett.* 83, 4713–4716 (1999).

[33] King, B. E., Wood, C. S., Myatt, C. J., Turchette, Q. A., Leibfried, D., Itano, W. M., Monroe, C., and Wineland, D. J. Cooling the collective motion of trapped ions to initialize a quantum register. *Phys. Rev. Lett.* 81, 1525–1528 (1998).

- On Page 5, “The train of laser pulses in the time domain can be considered as an equally spaced frequency “comb” [...] “, needs Reference.

We add the following reference:

[34] Diddams, S. A., Diels, J.-C., and Atherton, B. Differential intracavity phase spectroscopy and its application to a three-level system in samarium. *Phys. Rev. A* 58, 2252–2264 (1998).

- An example of less careful writing style: On Page 6, “The remaining steps are the standard sideband operations with [...]”, “[...] implemented by modulating the amplitude of the laser beams [...]”, and “By using the AWG, we are also able to add up signals by mathematical calculations at the sampling stage.”. What do the authors mean by “standard” and “amplitude of laser beams”, and what is the last sentence suppose to describe?

We improve the explanation for the experimental implementation with the modified Fig. 1. We rewrite the paragraph as follows.

“The second and third parts are realized by the red- and the blue-sideband transitions as shown in Fig.~\ref{fig:Displacement}(e). The time-dependent strength-coefficient $g(t)$ in Eq. (\ref{gt}) is implemented by the change of laser intensity, which is proportional to the RF power on the AOMs of Fig.~\ref{fig:Displacement}(a). We generate the time-dependent RF signal from an arbitrary waveform generator (AWG) and apply it to the AOM R2. By using the AWG, we can generate all the necessary RF frequencies and powers, which realizes the full Hamiltonian~(\ref{Hi}) containing the displacement operation, red- and blue-sideband transitions.”

- On Page 6, “In the experiment, we initialize the motional and internal state [...] by standard Doppler cooling, resolved sideband cooling and optical pumping.” needs a Reference.

We add the following two references:

[35] Heinzen, D. J. and Wineland, D. J. Quantum-limited cooling and detection of radio-frequency oscillations by laser-cooled ions. *Phys. Rev. A* 42, 2977–2994 (1990).

[36] Monroe, C., Meekhof, D. M., King, B. E., Itano, W. M., and Wineland, D. J. Demonstration of a fundamental quantum logic gate. *Phys. Rev. Lett.* 75, 4714–4717 (1995).

- An example, where experimental results are quoted without giving uncertainties: On Page 6, “The residual average phonon number and the heating rate are measured to be $\langle n \rangle = 0.016$ and $3.8 \text{ quanta s}^{-1}$), respectively.” Authors should indicate uncertainties, in particular, for non-specialists, who may not be able to judge significances of results for themselves.

We include the uncertainties for the residual average phonon number and heating rates in the main manuscript.

- On page 7, “Fig. 3(a) shows experimental data [...] quantitatively coincide with the theoretical calculations within experimental errors.” In general, in the Results part of the manuscript, it should be concisely stated what kind of

“theoretical calculations” have been performed, while in the Methods sections their details and limitations should be discussed. The “exact quantum dynamics”, on page 7, is not described in the methods section, while the description of the “20th-order perturbative calculations” is not sufficiently discussed. Why did the authors choose the 20th order? Can they visualize that perturbation calculations will not converge in the presented case (Fig. 3c) in comparison to cases where it does (e.g., Fig. 3b)? What kind of numerical calculation are performed to yield “exact quantum dynamics” and what are corresponding uncertainties and limitations for this?

As the reviewer suggested, we include the concise explanation for the theoretical calculation before the sentence starting with “Fig. 3(a) shows experimental data...” as follows.

“We compare the experimental results with the ideal theoretical calculations. In our simple situation of single bosonic, fermion and anti-fermion modes, we are able to numerically calculate the exact evolution with the full Hamiltonian. For the case that the exact numerical methods are not allowed as the system size grows, we also use a perturbative calculation discussed in Method section. ”

- On page 7, “Finally, as a demonstration of scalability, we realize fermion self-interaction processes extended to 2 bosonic modes by using both X and Y phonon modes of a single trapped ion.” While this is of course a first step in scaling the system, it should be carefully discussed what systematic effects come into play when increasing the number of participating modes. What kind of limitations are expected to emerge when increasing the number of modes even more? For example, how do the authors propose to significantly extract Fock state population distributions for an increasing number of modes? Why are experimental results for average number of virtual bosons in Fig. 3d missing?

For the quantum field theory, an interesting simulation would be the scattering of fermionic and anti-fermionic modes in continuous bosonic mode, which is closely related to real experimental situation. We may simulate the continuous regime of bosonic modes by increasing the number of phonon modes. In such simulation, the measurement of bosonic mode would be less important than that of fermionic or anti-fermionic mode. However, on the way of including large number of bosonic modes, we may have to measure the average number of each bosonic mode or the correlations among bosonic modes. In the revised version, we briefly introduce the method to measure the phonon numbers of multiple. We include these discussions in the last paragraph of the result as follows.

“By increasing the number of bosonic modes, we would simulate the continuous regime of bosonic modes, which would be related to scattering experiments. In such large number of bosonic modes, the non-perturbative behavior of fermionic or anti-fermionic mode could be intractable. On the way of increasing bosonic modes, a technology of correlation measurement of multiple phonon modes could be applied. [37,38]”

During the experiment, we had a technical difficulty to measure the average number of bosons, which as discussed above, is not directly related to the main message of the scalability. However, we note that there is no fundamental problem in measuring the average phonon numbers of multiple modes.

- On page 8, “ In particular, we remark that already with 10 two-level ions and 5 phononic levels per ion, one could perform quantum simulations of interacting quantum field modes that are beyond the reach of classical computations, that is, a Hilbert space dimension of $10^{10} \sim 2^{33}$, which would otherwise require a lengthy quantum algorithm with 33 qubits.” This statement is very strict and leaves no room for developments in efficient (classical) numerical approaches based on effective theories or models for the dynamics. Further, the authors should note that any complex simulation result has a verification problem as stated in criterion 5, discussed in Ref. [Cirac, J. I. & Zoller, P. Goals and opportunities in quantum simulation. Nat. Phys 8, 264–266 (2012)]. How would the authors suggest on dealing with this issue in their approach?

We point out that currently there are no known efficient classical algorithms for simulating interacting fermionic models in two or three spatial dimensions or for long-range couplings. With our algorithms, these models could be efficiently implementable in a trapped-ion quantum simulator. Perhaps this was not clear in the previous version, such that we have added, after the text “...which would otherwise require a lengthy quantum algorithm with 33 qubits.”, the paragraph

“We point out that there are no known efficient classical algorithms for simulating interacting fermionic models in arbitrary spatial dimensions, while with our approach, with a trapped-ion quantum simulator, fermionic models in arbitrary dimensions could be analyzed with polynomial resources.”

Regarding the verification of the outcome of the quantum simulation, we point out that in our scattering model all that one would need to measure, to detect either bosonic or fermion-antifermion pair creation, would be, respectively, the average number of bosons or the population of the respective internal level. Therefore, the measurement would be efficient with current trapped-ion technology. To clarify this further, we added, after “...while with our approach, with a trapped-ion quantum simulator, fermionic models in arbitrary dimensions could be analyzed with polynomial resources.”, the sentence

“The verification of our proposed scalable experiment requires polynomial resources, as for the detection of the number of bosonic excitations produced, or the population of the fermionic or antifermionic states, only a polynomial number of measurements is required.”

- In the Methods section, relevant details on the experimental techniques are not cited properly and uncertainties are not discussed, in particular, in the section about “Fermionic state measurement” and “Average boson number measurement”. Authors should cite original work, which introduced used methods, and they should discuss statistical as well as systematic uncertainties

in their experiment. In particular, it is not valid to just write the following: "After careful beam alignment and quarter wave plate adjustment, the measured strength coefficients of [...] are $(2\pi)7.2$ kHz and $(2\pi)14.4$ kHz, respectively, which are consistent with the theory ratio 1: 2." How large are the uncertainties on the experimental rates and how would a deviation from that 'theory ratio' show up in their results?

In the Method section of "Fermionic state measurement," we include the Refs. [39-41] for the phonon projective measurement. And we briefly explain the source of the uncertainty as follows in the section.

"The uncertainty of the measurement mainly comes from the quantum projection noise of binary result of single measurements."

In the Method section of "Average boson number measurement," we include the proper original references. The uncertainty for this measurement is explained as follows.

"The main uncertainty in the average phonon measurement comes from fitting and we include one standard deviation as an uncertainty throughout the manuscript."

Reviewer #3 (Remarks to the Author):

In their manuscript, Zhang and coworkers report on a trapped ion experiment, where a generalized Jaynes-Cummings model is realized using four internal states of a trapped Yt ion, coupled to vibrational modes. The experiment follows closely previous proposals, in particular Ref. [23], where the microscopic Hamiltonian is mapped onto a minimal field theory made of one particle with fermionic statistics, one anti-particle, and bosonic fields. The main difference with respect to the initial proposal seems to be that here a single ion is used, instead of two, mapping the four available fermionic states onto 4 internal states. After an initial state preparation of the system, the authors let the system evolve (coherently, given the relatively slow dissipation rate) and perform measurements of the state populations of both fermions and bosons, and some basic correlation functions. They observe significant deviations from both the experiment and exact numerics with respect to perturbation theory - the latter cannot capture the system dynamics after long times.

We are very glad to know that the Reviewer acknowledges the breakdown of the theoretical perturbation theory, à la Feynman, when compared to our experimental results. This is one of the key justifications of the validity and scalability aspects of our quantum simulation approach.

From the experimental viewpoint, the work is sound. Several details are provided, and the prescription to load the initial state and perform the time evolution are well explained. From the theoretical viewpoint, the mapping from Eq. (6) to the ion and bosonic modes is clearly explained. What is less clear is the mapping from Eq. (1) to Eq. (2-3) - in particular, one wonders why a time-

independent Hamiltonian operator is mapped onto a time-dependent one. This seems to be covered in Ref. [23]; nevertheless, a qualitative explanation on how this actually works would be highly desirable even in the present manuscript.

We agree with the Reviewer in that our formalism, detailed in Ref. [23], is missing some clarifications in describing the experimental implementation. Therefore, we added a paragraph, after the text "T is the total time of the pair-creation process while σt is the temporal interval of the interaction region." following Eq. (3):

"Our formalism, explained in detail in Ref. [23], involves considering incoming comoving fermionic and antifermionic modes at lowest order in perturbation theory. The time dependence of the interaction of the incoming particles, as they collide, maps onto a time dependence of the interaction Hamiltonian coupling."

While I believe the work reported is valid, in my opinion, its results do not match the novelty and impact criteria of Nature Communications. Let me detail below my criticisms.

We also answer in detail to each raised issue below.

1) The experiments simulate the physics of a rather simple model Hamiltonian. In this respect, this work constitutes a natural follow up of the remarkable simulation of the Dirac equation proposed and realized by some of the authors in Refs. [10,11].

We appreciate the comment of the Reviewer concerning the pioneering previous theoretical and experimental works in the quantum simulation of the Dirac equation in first quantization. But we would like to point out that such a model of the Dirac equation has a hard time when scalability is in the spot, even if it is still possible to go up towards models involving 2+1 and 3+1 dimensions. On the contrary, in our submitted manuscript, it is highly motivating and novel to deal with a quantum field theory model, in second quantization, with all the flesh and subtleties associated to it.

The main novel aspect here is the additional coupling to bosonic modes. However, this is rather different from what naturally occurs in quantum field theories, where the bosonic degrees of freedom are gauge couplings - as reported, e.g., in Ref. [21], consisting of two pairs of particles and antiparticles.

We do appreciate the experimental work and efforts described in Ref. [21]. However, in our quantum simulation of fermionic scattering, it is crucial to explicitly involve the bosonic degrees of freedom. These are not meant to be integrated or traced out from the model, they are describing real bosonic excitations that mediate the fermionic coupling.

More importantly, I think the statement made by the authors, that their work is the first experimental realization of a trapped ion system simulating the dynamics of Hamiltonians involving both fermions and bosons, is not correct -

this was done, including gauge symmetries, already in Ref. [21]. This somehow severely mitigates the novelty and impact of the present work - there have been already building block experiments showing how trapped ions can indeed serve as quantum simulators of quantum field theories including both fermions and (gauge) bosons.

With all our deep respect, we are puzzled by this comment of the Reviewer in the light of his previous comment, where he/she mentions Ref. [21] to point out that some quantum field theory models do not need to explicitly describe bosonic degrees of freedom. In consequence, in that reference, neither bosons explicitly appear in the simulated model nor in the simulating experiment. This is a rather obvious observation given that the 4 ion qubits used in Ref. [21] can only describe 16 dimensions, and no infinite-dimensional bosonic degree of freedom could be considered in general. In our experiment, we do use not only one but two bosonic modes of the simulated model encoded in two motional modes, apart from the additional internal ionic degrees of freedom. Therefore, we keep our evident claim that our experiment is the first one simulating a quantum field theory model in trapped ions, where we consider bosonic degrees of freedom in the simulated model and also in the simulating system.

2) Given the previous consideration, one could wonder whether the present scheme can nevertheless be useful in understanding some not-easy-to-simulate physical phenomena. The authors provide a simple demonstration that this could potentially be the case, by showing that the system can cope well with strong coupling - a regime where controlled numerics are challenging. However, the main question one has is how well this system is scalable, and what questions can really be answered in the close future.

Regarding scalability, as mentioned to the Editor and the first two Reviewers, we point out to the two PRL articles from some of us, and we have included the paragraph at the end of page 2:

"The scalability of our protocols for simulating fermionic models in two and three spatial dimensions was demonstrated in Ref. [24]. There, it is shown that the many-body operators appearing after mapping of the fermionic modes onto spins via Jordan-Wigner transformation, can be efficiently implemented via a combination of two Mersenne-Stern gates and a local gate. We also proved in Ref. [25] that one may add bosonic modes to the fermionic simulation in a scalable and efficient way, by means of a digital-analog quantum simulator.", where the new Ref. [25] is the paper by Mezzacapo et al., Phys. Rev. Lett. 109, 200501 (2012), and all previous references are therefore shifted one number.

2a - Scalability of the approach is not discussed at all. In particular, a detailed feasibility study should be provided to demonstrate that larger systems can be used to faithfully reproduce the target dynamics. This point is critical, in particular given the fact that it would be hard to control at the same time a relatively large amount of couplings following the present scheme. This even

without considering the fact that, at present, only one-dimensional systems can be accessed with reasonable tunability and fidelities.

We point out that, with a 1-dimensional ion chain, one can efficiently simulate fermionic modes in arbitrary spatial dimensions by means of Mersersengren gates and local gates, as we proved in Ref. [24]. By means of a digital-analog quantum simulator, as exposed in our previous paragraph in this reply, one may also add bosonic modes in a scalable way, with full control of the different couplings involved, because of the digital component of the simulator.

2b - Another important point which is not discussed is whether the present approach is opening new doors to solve unsettled problems. In the conclusions, the authors state that considering 10-ions systems, they could tackle problems at the boundary of exact diagonalization. (one could argue that this is not beyond classical computations, as matrix-product state methods can handle relatively easily the time evolution of such small one-dimensional systems, for the case of field theories see, e.g., Banuls et al., POS (Lattice 2013) 332). However, it is unclear which open, relevant problem can actually be answered in this context - what is somebody ultimately learning by these experiments if they can indeed be scaled up to 10 ions?

We point out that fermions, or fermions coupled to bosons, cannot be efficiently simulated with matrix-product states or any other classical numerical method in two or three spatial dimensions. Otherwise, scalable quantum simulations of models involving spins, fermions, and bosons, would not make any sense as a quantum technology, which is certainly not the case. With our protocols, as demonstrated in Ref. [24] and Ref. [25] in the new version of the paper, we can address these important situations with an efficient digital-analog quantum simulator that employs only polynomial resources.

In view of these last two criticisms, the impact of the present paper is very unclear, especially outside of the trapped ion community.

With our deep respect, we disagree with this assertion. We believe that, while praising the immense efforts of previous works, we have given strong arguments above to keep the central claims of our pioneering experimental results.

In summary, I believe this is valid work, but it does not meet Nature Communications high standards of impact and novelty. I have few additional comments that the authors might find useful addressing for future submissions:

a) the word 'antifermonic' is rather uncommon. While its meaning is clear within the present context, I'd suggest to change wording, and refer to the fermionic modes as particle and antiparticle.

The word "antifermion" is extensively used in the high-energy physics community, and "antifermonic" is just the adjectivation of the previous one, which is also employed in the literature. Therefore, we prefer to keep this denominations in our manuscript.

b) for quantum simulators to approach field theories, one typically requires Lorentz-invariance to be recovered at least in some limits. How can one see this aspect in the present implementation?

Our formalism can reproduce the relativistic field theory Hamiltonian term by term, via the digitization of the dynamics. Therefore, it can implement the Lorentz covariance of the theory by suitable quantum control of the couplings to emulate the fermionic Dirac spinor structure, photonic polarizations, and relativistic energy dispersion.

c) in the pdf version that I got, there are no experimental points in Fig. 3d for both the green and blue curves, not sure why.

We did not measure the average phonon numbers of both modes due to technical difficulties. Eventually, however, it is not critical in this manuscript since the main interest would be in the dynamics of fermionic and anti-fermionic mode for the large size of bosonic modes.

Reviewers' comments:

Reviewer #1 (Remarks to the Author):

I am traveling, but was able to look over the referee responses to my comments. In my view they have done a sufficient job responding to my comments (reviewer 1), and I believe the manuscript ready for publication in Nature Communications.

Unfortunately I am not able in a timely manner to dissect the other reviewers comments and the referee responses.

Reviewer #2 (Remarks to the Author):

May, 24th 2017

In this first revision of the manuscript of X. Zhang et al, entitled "Fermion-antifermion scattering via boson exchange in a trapped ion", the authors sufficiently address only some of my previous concerns.

In particular, the manuscript still lacks a concise description of calculations that are supposed to validate experimental results. Further, a critical discussion about systematic effects of the particular experimental approach is needed. In combination, the authors fail to present sufficient evidence for their drawn conclusions. Adding missing details to the manuscript is essential to reach a level that enables other researchers to reproduce and build on the authors intriguing work. In addition, it seems that authors introduce careless mistakes during their first revision.

In conclusion, I am not able to endorse publication at this stage.

In the case of a second revision, authors may address my remaining comments and concerns:

- The issue of scalability is not sufficiently answered: I acknowledge that it has been shown (theoretically) in previous work, that simulations can be “efficiently” performed in a scalable way. But, I would like to stress the fact that uncertainties and inaccuracies due to systematic and statistical limitations will increase when increasing the complexity of the experimental quantum simulation. How would the authors experimentally implement a further increased number of fermionic or bosonic modes in their approach? What kind of experimental limitations are expected to emerge when increasing the number of modes?

- What kind of “technical difficulty” prevented the authors from measuring the average Fock-state numbers in the case shown in Fig. 3d? The authors write: “We choose the initial state to contain one fermion and no bosons [...]”. How did the authors ensure that they prepared both modes close to the ground state if they were not able to determine the average Fock-state numbers? In addition, details are missing on how the authors experimentally tune the coupling to both radial modes. How much is the coupling to mode Y suppressed, in the cases shown in Fig. 3a-c?

- Further, I am concerned about the missing description of numerical simulations that are supposed to verify the experimental results in their manuscript. So far, the authors write: “We compare the experimental results with the ideal theoretical calculations. In our simple situation of single bosonic, fermion and anti-fermion modes, we are able to numerically calculate the exact evolution with the full Hamiltonian. For the case that the exact numerical methods are not allowed as the system size grows [...]” The authors need to outline their procedures in detail for the “ideal theoretical calculations” in the Methods section, addressing the questions: How are numerical calculation performed to yield “exact evolution with the full Hamiltonian”, how many (free) parameters does it contain, how did they set them, and what are corresponding uncertainties and limitations of these simulations? How do the authors define 'not allowed'? At what size or parameter regimes do numerical calculations become unable to verify experimental results due to significant inaccuracy?

- A related issue is that the description of the “Feynman diagram calculation” is still not detailed: Why did the authors choose the 20th (and $m = 100$) order? What kind of formal criterion and how much effort did they choose, when cutting off at the 20th order? Plots showing the convergence (or the missing convergence) for the different regimes are needed and would be helpful to illustrate such criterion. Such discussion is required to introduce non-specialists to such calculations and to enable a fair judgment of the authors' claims by the reader.

A few examples of minor issues and careless mistakes in the manuscript are:

- Figure caption 1: “[...], which is represented in the upper equation.” To what equation do the authors refer?

- In Figure 2 (and caption), symbols “f” are not consistent with the main text. Note, throughout the manuscript several symbols are used ambiguously, e.g., “n” and “m”. Further, the use of indices is not consistent. For example, in case of single bosonic mode, authors use “n” (without index) and “a₀”, while in the case of two bosonic modes, they use “n₁ and n₂” and “a₁ and a₂”

- Figure 3: Panel b) seems to be a copy of panel a)--described data is not shown in b), but was presented in the previous version of the manuscript.

- In the revision, authors add: “The uncertainty of the measurement mainly comes from the quantum projection noise of binary result of single measurements.” The term “quantum projection noise” needs a reference, as it is not commonly used in all potentially interested communities.

Reviewer #3 (Remarks to the Author):

In their resubmission, the authors clarified some of the points I raised in my previous report.

However, I disagree with some of the major comments:

1) the point I disagree upon is quite simple: in order to simulate a theory with fermions and bosons, one does **not** have to realize both degrees of freedom.

In particular, a quantum simulation or only, say, bosons (or by any case, a classical one) can address a certain quantum problem that contains bosons and fermions.

This is also what is performed in the actual reported experiment (which, let me remark again, it technically impressive): there are no microscopic fermions, but the latter degrees of freedom are still smartly encoded in dynamics of the ions and the phonons.

This is the same also in classical simulations: for example, in certain lattice field theory methods (see e.g. the work in the Munich group), only bosons or spins are encoded, but still this gives concrete predictions for models including bosons and fermions.

Given this, I still sharply disagree with the authors, that they report the first quantum simulation of a quantum field theory including bosons and fermions.

2a) I thank the authors for the clarification and appreciated the amendments.

2b) I'm not sure I agree with the authors here. In order to substantiate their claim, the authors should indicate a concrete problem, and not 'important situations'. This statement is too vague and leaves the impression that it is unclear if possible at all. The authors should state (a) a concrete, open problem which is inaccessible to present day approaches, and (b) explicitly mention the resources (number of ions and dimensionality) needed to address it. This perspective is extremely important to judge whether this approach can make an impact in the close-to-mid term.

On my minor comments, I still find unfit the choice of the word 'antifermion', I'm satisfied with the author responses.

Response to Reviewer #2

In particular, the manuscript still lacks a concise description of calculations that are supposed to validate experimental results. Further, a critical discussion about systematic effects of the particular experimental approach is needed. In combination, the authors fail to present sufficient evidence for their drawn conclusions. Adding missing details to the manuscript is essential to reach a level that enables other researchers to reproduce and build on the authors intriguing work. In addition, it seems that authors introduce careless mistakes during their first revision.

We greatly appreciate that the reviewer has requested a more detailed explanation of the calculations related to the experiment. As we will discuss later in more detail, during our revision we have improved in the resubmitted manuscript the techniques used for the perturbative calculations. In addition, we have overcome previous experimental difficulties and include new experimental data concerning measurements of the two vibrational bosonic modes, as requested. Furthermore, we also checked the manuscript in detail and believe that its quality is suitable for publication in Nature Communications.

In the case of a second revision, authors may address my remaining comments and concerns:

- The issue of scalability is not sufficiently answered: I acknowledge that it has been shown (theoretically) in previous work, that simulations can be “efficiently” performed in a scalable way. But, I would like to stress the fact that uncertainties and inaccuracies due to systematic and statistical limitations will increase when increasing the complexity of the experimental quantum simulation. How would the authors experimentally implement a further increased number of fermionic or bosonic modes in their approach? What kind of experimental limitations are expected to emerge when increasing the number of modes?

In the resubmitted manuscript, we discuss the systematic effects and limitations further, in the light of our experimental demonstration. Along these lines, we have added the following explanatory text:

“In conclusion, this work considers an experimental quantum simulation of interacting fermionic and bosonic quantum field modes. Our approach can be scaled up by progressively incorporating more fermionic and bosonic field modes, which may lead to a full-fledged digital-analog quantum simulation of quantum field theories such as QED \cite{Peskin, Casanova11b, Casanova12} or the Holstein model \cite{Mezzacapo12}, where correlations between multiple fermions and phonons have critical relevance. In our current experimental system, an extension to multi-fermion and multi-phonon (bosonic) modes should be straightforward by loading a number of ions in a single trap, where the spins of ions map the fermionic modes through Jordan-Wigner transformation and the vibrational modes of ions directly map the bosonic modes. Other than spin-spin interactions in the Holstein model, for example, the couplings between fermionic modes and bosonic modes can be implemented by the same Raman laser beams that are individually addressing single ions and tuned to specific mode frequencies. And it has been shown that the number of gates grows polynomially as the number of fermions and bosons \cite{Mezzacapo12}. The Ref. \cite{Mezzacapo12} discussed the estimated infidelities from the gate errors in realistic experimental decoherence condition up to four sites, which clearly showed the degree of control is more demanding when the coupling strengths between the modes increase. As demonstrated in our experiment, we do not observe any obvious degradation of the

simulation when using two modes, though here we do not have the technical problem of individual addressing. We may implement the model in a fully analogue way together with proper spin-spin interactions \cite{porras2004effective, kim2008geometric, kim2009entanglement}, which would allow us to study the pairing or polaron physics occurring in many unconventional superconducting systems \cite{hague2012bilayers, stojanovic2012quantum} with the controls of various parameters. In particular, we remark that already with 10 two-level ions and 5 phononic levels per ion, one could perform quantum simulations of interacting quantum field modes that are beyond the reach of classical computations, that is, a Hilbert space dimension of $10^{10} \sim 2^{33}$, which would otherwise require a lengthy quantum algorithm with 33 qubits. This experiment opens an avenue that aims at outperforming the limitations of classical computers, with in principle scalable quantum simulations.”

- What kind of “technical difficulty” prevented the authors from measuring the average Fock-state numbers in the case shown in Fig. 3d? The authors write: “We choose the initial state to contain one fermion and no bosons [...]”. How did the authors ensure that they prepared both modes close to the ground state if they were not able to determine the average Fock-state numbers? In addition, details are missing on how the authors experimentally tune the coupling to both radial modes. How much is the coupling to mode Y suppressed, in the cases shown in Fig. 3a-c?

The main technical difficulty in our experiment was that the heating of the mode Y is much more severe than that of the mode X, though we are able to cool down both modes to the ground states. Kuan Zhang and Shuaining Zhang (who is now included in the author list) resolved the technical problem and now we are glad to present the new data for both modes. As shown in Fig. 3d, the experimental results and theoretical expectation are in good agreement and we do not see a serious limitation up to two motional modes in our analogue simulation. We include the description of the new data for Fig. 3d as follows.

“We also clearly observe the dynamics of both bosonic modes in a good agreement with the theoretical expectation.”

In the experiment, the secular frequencies of the X and Y modes are $(2\pi) (2.4, 1.9)$ MHz, respectively and most of the experiments were performed using the X mode. The typical coupling strength g_1/ω_0 of the X mode is around 0.1, where $\omega_0 \sim (2\pi)10$ kHz. Since the detuning of the laser beam to the mode Y is $(2\pi) 490$ kHz (49 times larger than ω_0), the effect of the laser beams coupling to the mode Y is also 49 times smaller, that is, $g_2/[(2\pi) 490 \text{ kHz}] = 1/490 \sim 0.002$, which cannot make any observable effect in the experiment.

We briefly include the above discussion in the last paragraph of the section “Results” as follows.

“We set $g_1=0.15\omega_0$, the first boson mode frequency $\omega_1=\omega_0$ and the second boson mode frequency $\omega_2=0.9\omega_0$. We note that the g_1 (g_2) coupling to the mode Y (X) is negligible since the detuning to the mode Y (X) is larger by a factor of 50, which effectively suppresses the strength by the same amount.”

- Further, I am concerned about the missing description of numerical simulations that are supposed to verify the experimental results in their manuscript. So far, the authors write: “We

compare the experimental results with the ideal theoretical calculations. In our simple situation of single bosonic, fermion and anti-fermion modes, we are able to numerically calculate the exact evolution with the full Hamiltonian. For the case that the exact numerical methods are not allowed as the system size grows [...].” The authors need to outline their procedures in detail for the “ideal theoretical calculations” in the Methods section, addressing the questions: How are numerical calculation performed to yield “exact evolution with the full Hamiltonian”, how many (free) parameters does it contain, how did they set them, and what are corresponding uncertainties and limitations of these simulations? How do the authors define 'not allowed'? At what size or parameter regimes do numerical calculations become unable to verify experimental results due to significant inaccuracy?

We simply use the built-in function in Mathematica for the numerical calculations. We include the “ideal theoretical calculations” in the Methods section as the reviewer suggested:

“The exact dynamics of the Hamiltonian (\ref{Hi}) can be obtained by solving the time-dependent Schrödinger equation $i \hbar \frac{\partial}{\partial t} \ket{\psi(t)} = H_{\{I\}}(t) \ket{\psi(t)}$. We numerically solve the equation with the built-in function of Mathematica, ‘NDSolve’, which finds a numerical solution to the ordinary differential equation mainly based on Runge-Kutta method. In the numerical calculation, we include 4 internal levels and up to 10 phonons per mode for the Hamiltonian (\ref{Hi}), which changes the Schrödinger equation to the ordinary differential equation with 40 and 400 components for single mode and two modes of the state $\ket{\psi}$, respectively. With the option of infinite \$Maxsteps\$ in Mathematica, the numerical calculations converge and do not show any error messages. We note that there is no free parameter except the initial condition of $\ket{0_f 0_f}$ and the ground state of modes. The main limitation of the numerical calculation would be the size of the Hilbert space when we scale up the system with multiple fermions and bosons.”

- A related issue is that the description of the “Feynman diagram calculation” is still not detailed: Why did the authors choose the 20th (and $m = 100$) order? What kind of formal criterion and how much effort did they choose, when cutting off at the 20th order? Plots showing the convergence (or the missing convergence) for the different regimes are needed and would be helpful to illustrate such criterion. Such discussion is required to introduce non-specialists to such calculations and to enable a fair judgment of the authors' claims by the reader.

We greatly appreciate the reviewer to raise such detailed questions for our “Feynman diagram calculation.” Actually, we have now improved the techniques and readability of that part, as newly described in the Methods section.

We include the result of new perturbation method in Fig. 3c, which shows the idea of convergence of the perturbation. We believe the new graph in Fig. 3c, Fig. 4 and the methods section provides the evidence to enable non-specialists to judge the validation of our claim.

Accordingly, we modify the caption of Fig. 3c as follows.

“Solid lines are obtained by exact numerical simulation using the built-in solver of the ordinary differential equation in Mathematica. Dashed lines are computed by a Dyson series expansion with Feynman diagrams up to 1st and 3rd orders after dividing the whole time by 100 (see Methods). By including the Dyson series up to 7th order, the errors from the exact numerical calculation are below 10^{-7} (see Methods and Fig. 4)”

- Figure caption 1: “[...], which is represented in the upper equation.” To what equation do the authors refer?

Originally, we referred to the equation in the Fig. 1a. In order to avoid the confusion, we removed the sentence.

- In Figure 2 (and caption), symbols “ f ” are not consistent with the main text. Note, throughout the manuscript several symbols are used ambiguously, e.g., “ n ” and “ m ”. Further, the use of indices is not consistent. For example, in case of single bosonic mode, authors use “ n ” (without index) and “ a_0 ”, while in the case of two bosonic modes, they use “ n_1 and n_2 ” and “ a_1 and a_2 ”

We understand that the symbols “ f ” for the frequencies in the Figure 2 (and caption) can be confused with “ f ” for the fermion in the main text. We changed “ f ” for the frequency to “ ω ” to avoid such confusion and to make it consistent with the main text.

- Figure 3: Panel b) seems to be a copy of panel a)---described data is not shown in b), but was presented in the previous version of the manuscript.

We have corrected it in the revision.

- In the revision, authors add: “The uncertainty of the measurement mainly comes from the quantum projection noise of binary result of single measurements.” The term “quantum projection noise” needs a reference, as it is not commonly used in all potentially interested communities.

In a single measurement, we obtain only the binary results, fluorescence or no fluorescence from atom. After taking multiple measurements, we obtain the average value and the standard deviation, which we call the “quantum projection noise.” We include the following reference as suggested.

[48] Itano, W. M., Bergquist, J. C., Bollinger, J. J., Gilligan, J. M., Heinzen, D. J., Moore, F. L., Raizen, M. G., and Wineland, D. J., Quantum Projection Noise: Population Fluctuations in 2-Level Systems Phys. Rev. A **47**, 3554–3570 (1993).

Response to Reviewer #3

1) the point I disagree upon is quite simple: in order to simulate a theory with fermions and bosons, one does *not* have to realize both degrees of freedom. In particular, a quantum simulation or only, say, bosons (or by any case, a classical one) can address a certain quantum problem that contains bosons and fermions.

This is also what is performed in the actual reported experiment (which, let me remark again, it technically impressive): there are no microscopic fermions, but the latter degrees of freedom are still smartly encoded in dynamics of the ions and the phonons.

This is the same also in classical simulations: for example, in certain lattice field theory methods (see e.g. the work in the Munich group), only bosons or spins are encoded, but still this gives concrete predictions for models including bosons and fermions.

Given this, I still sharply disagree with the authors, that they report the first quantum simulation of a quantum field theory including bosons and fermions.

We agree with the referee and retire the claim. At the same time, we are glad to read that the Referee considers our experiments as “impressive”.

2b) I’m not sure I agree with the authors here. In order to substantiate their claim, the authors should indicate a concrete problem, and not ‘important situations’. This statement is too vague and leaves the impression that it is unclear if possible at all. The authors should state (a) a concrete, open problem which is inaccessible to present day approaches, and (b) explicitly mention the resources (number of ions and dimensionality) needed to address it. This perspective is extremely important to judge whether this approach can make an impact in the close-to-mid term.

In the Ref. [24] and Ref. [25], concrete, open problems, which are inaccessible to current approaches, have been seriously and carefully discussed including explicit resources. This is also a similar request from the Reviewer #2, and accordingly we revised the section of Discussion to state concrete, open problems with explicitly mentioning the resources as follows.

“In conclusion, this work considers an experimental quantum simulation of interacting fermionic and bosonic quantum field modes. Our approach can be scaled up by progressively incorporating more fermionic and bosonic field modes, which may lead to a full-fledged digital-analog quantum simulation of quantum field theories such as QED \cite{Peskin, Casanova11b, Casanova12} or the Holstein model \cite{Mezzacapo12}, where correlations between multiple fermions and phonons have critical relevance. In our current experimental system, an extension to multi-fermion and multi-phonon (bosonic) modes should be straightforward by loading a number of ions in a single trap, where the spins of ions map the fermionic modes through Jordan-Wigner transformation and the vibrational modes of ions directly map the bosonic modes. Other than spin-spin interactions in the Holstein model, for example, the couplings between fermionic modes and bosonic modes can be implemented by the same Raman laser beams that are individually addressing single ions and tuned to specific mode frequencies. And it has been shown that the number of gates grows polynomially as the number of fermions and bosons \cite{Mezzacapo12}. The Ref. \cite{Mezzacapo12} discussed the estimated infidelities from the gate errors in realistic experimental decoherence condition up to four sites, which clearly showed the degree of control is more demanding when the coupling strengths between the modes increase. As demonstrated in our experiment, we do not observe any obvious degradation of the simulation when using two modes, though here we do not have the technical problem of individual addressing. We may implement the model in a fully analogue way together with proper spin-spin interactions \cite{porras2004effective, kim2008geometric, kim2009entanglement}, which would allow us to study the pairing or polaron physics occurring in many unconventional superconducting systems \cite{hague2012bilayers, stojanovic2012quantum} with the controls of various parameters. In particular, we remark that already with 10 two-level ions and 5 phononic levels per ion, one could perform quantum simulations of interacting quantum field modes that are beyond the reach of classical computations, that is, a Hilbert space dimension of $10^{10} \sim 2^{33}$, which would otherwise require a lengthy quantum algorithm with 33 qubits. This experiment opens an avenue that aims at outperforming the limitations of classical computers, with in principle scalable quantum simulations.”

Reviewers' comments:

Reviewer #2 (Remarks to the Author):

September, 29th 2017

In this second revision of the manuscript of X. Zhang et al, entitled "Fermion-antifermion scattering via boson exchange in a trapped ion", the authors sufficiently address some of my previous concerns and I highly appreciate their effort to explain more crucial details of their work.

The authors were required to reconsider some of their calculations and experimental results and present new graphs and data that present, in parts, substantially different results as prior to their revision. This suggests (severe) mistakes within the earlier versions and authors neither discuss details of these changes nor elaborate on the impact of their conclusions. In this context, it seems that authors introduce (again) careless mistakes in the manuscript. In combination, the authors miss to convince me that they have "checked the manuscript in detail and [...] that its quality is suitable for publication" and it raises some doubts about the scientific validity of their work.

In conclusion, unfortunately, I am still not able to recommend publication.

In case of a third revision, authors need to most carefully address the following comments and concerns:

After considering to introduce a formal criterion for convergence of the Dyson series, the authors found, as depicted in Fig. 4 and discussed in the corresponding caption, that the seventh order is able to describe their results in Figure 3c. But in the main text, they still write: "Next, we realize a nonperturbative process of scattering [...]. Fig. 3(c) shows the quantum dynamics of this strong coupling situation, where we cannot easily discriminate the contributions from the self-interaction and pair production processes. Moreover, while the experimental data coincide with the exact numerical calculation, we observe huge deviations between the exact quantum dynamics and the 20th-order perturbative calculations (see Methods), which confirms the fact that we simulate the dynamics in the nonperturbative regime.". This is obviously inconsistent (also seen in Fig 3c) and authors need to carefully clarify this and outline the corresponding impact on their drawn conclusions. In the other cases presented in Fig. 3, the second or, even, the first order is sufficient to

reproduce ideal calculations with high fidelity. Why are these numerical results substantially different than in the previous versions, where the authors claimed that the 20th order describes results in Fig. 3a,b, and d, and cannot explain data in Fig. 3c?

In this context, the authors write in the caption of Fig. 3: “By including the Dyson series up to the 7th order, the errors from the exact numerical calculation below 10^{-7} ”. Here, the authors may want to write “deviations” instead of “errors”.

Coming back to the previous questions: “How would the authors experimentally implement a further increased number of fermionic or bosonic modes in their approach? What kind of experimental limitations are expected to emerge when increasing the number of modes?” In particular, how well defined remain the couplings when having ten ions with a denser motional mode spectrum? In addition, authors now write: “As demonstrated in our experiment, we do not observe any obvious degradation of the simulation when using two modes”. I acknowledge that for a single ion, modes are well separated, but this separation substantially reduces with increasing number of ions. Further, I assume that following the approach, one needs to ensure that ion crystals remain in linear order, which will eventually require that axial frequencies need to be reduced and cooling becomes less efficient, heating rates higher, and the mode-spectrum density increases even further. How does the experimental effort scales with the number of constituent fermions and bosons? The authors need to make a fair statement about these limitations, although these limits may not be of fundamental character.

Concerning the description of the “exact dynamics” calculation: Authors write that “We note that there is no free parameter except the initial condition of $|\text{ket}\{0_f 0_f\}$ and the ground state of modes”. What do the authors consider to be free parameters? The number of modes, coupling strength, detunings, etc. are also relevant parameters! Did the authors set values to all of these parameters according to experimentally determined parameters, or did they make any kind of numerical model fit to the data? How do experimental parameter uncertainties translate into numerical confidence intervals? The authors still write in the main text: “In our simple situation of single bosonic, fermion and anti-fermion modes, we are able to numerically calculate the exact evolution with the full Hamiltonian. For the case that the exact numerical methods are not allowed as the system size grows [...]” And, they did not reply to the earlier questions: “How do the authors define 'not allowed'? At what size or parameter regimes do numerical calculations become unable to verify experimental results due to significant inaccuracy? What kind of formal quality factor did the authors consider in this case? Note, all of these questions need to be carefully clarified, in order to enable a fair judgment of the scientific quality of the results.

Reviewer #3 (Remarks to the Author):

The authors amended the manuscript and replied to all my previous comments. While I am still not fully convinced about the overall suitability of the paper for Nature Communications, after considering the positive opinions of my peers, I now suggest publication of the work as is.

Reply to Reviewer #2

In this second revision of the manuscript of X. Zhang et al, entitled “Fermion-antifermion scattering via boson exchange in a trapped ion”, the authors sufficiently address some of my previous concerns and I highly appreciate their effort to explain more crucial details of their work.

We also greatly appreciate the Reviewer #2 for the serious and persistent reviews, which have allowed us to improve the quality of the paper significantly.

After considering to introduce a formal criterion for convergence of the Dyson series, the authors found, as depicted in Fig. 4 and discussed in the corresponding caption, that the seventh order is able to describe their results in Figure 3c. But in the main text, they still write: “Next, we realize a nonperturbative process of scattering [...]. Fig. 3(c) shows the quantum dynamics of this strong coupling situation, where we cannot easily discriminate the contributions from the self-interaction and pair production processes. Moreover, while the experimental data coincide with the exact numerical calculation, we observe huge deviations between the exact quantum dynamics and the 20th-order perturbative calculations (see Methods), which confirms the fact that we simulate the dynamics in the nonperturbative regime.”. This is obviously inconsistent (also seen in Fig 3c) and authors need to carefully clarify this and outline the corresponding impact on their drawn conclusions.

We agree with the Reviewer, this was a misprint we made when writing the previous version. We have now revised the paragraph related to the result in Figure 3c as follows,

“Next, we realize the process of scattering with parameters $g_{1}=0.1\omega_{0}, g_{2}=\omega_{0}, \sigma_{t}=4/\omega_{0}$, where $g_{2}\geq \omega_{0}$. In this regime, the interaction Hamiltonian (\ref{HInter}) cannot be regarded as a perturbation. In such a strong coupling situation, we cannot easily discriminate the contributions from the self-interaction and pair production processes. When the initial fermion-antifermion pair disappears, more than a single boson is created in the process as shown in Fig. \ref{fig:TwoModes}(c), which is qualitatively different from the dynamics shown in Fig. \ref{fig:TwoModes}(b). Since the size of the Hilbert space is not too large, we numerically calculate the dynamics of the Hamiltonian by direct numerical integration (see Methods), which is in agreement with the experimental results shown in Fig. \ref{fig:TwoModes}(c). However, as the number of fermion-antifermion pairs and bosons increases, the exact numerical calculation will be intractable by classical means. We have also developed a perturbation method based on the observation that, for a reasonably small time $g_{2} t \ll 1$, the effect of the coupling term g_{2} does not produce a divergence in the dynamics. We divide the total time of the process by 100 and apply the perturbation method (see Methods) to the unitary evolution operator in each time slice. We find that after including terms up to the 7th order in the perturbation parameter, the deviation of the perturbative dynamics from the complete one is below 10^{-7} . However, even this approach, based on a perturbative expansion within time slices, would be difficult to use for large Hilbert space dimensions with more fermions and bosons.”

In the other cases presented in Fig. 3, the second or, even, the first order is sufficient to reproduce ideal calculations with high fidelity. Why are these numerical results substantially different than in the previous versions, where the authors claimed that the 20th order describes results in Fig. 3a,b, and d, and cannot explain data in Fig. 3c?

Figures 3a,b, and d in the previous version of the manuscript were performed with an improved analysis in perturbation theory with respect to the initially submitted manuscript. The reason for this is that we found an inconsistency of the theoretical analysis in the initial version of the paper. This might have been confusing, as the Reviewer points out, given that in the previous resubmission we included a misprint when referring to the 20th order of the initial version. We apologize for that, since in the previous version of the manuscript the 20th order was, obviously, not anymore applicable. The version now submitted is consistent and we have fully corrected this issue.

In this context, the authors write in the caption of Fig. 3: “By including the Dyson series up to the 7th order, the errors from the exact numerical calculation below 10^{-7} ”. Here, the authors may want to write “deviations” instead of “errors”.

We have changed the term as suggested.

Coming back to the previous questions: “How would the authors experimentally implement a further increased number of fermionic or bosonic modes in their approach? What kind of experimental limitations are expected to emerge when increasing the number of modes?” In particular, how well defined remain the couplings when having ten ions with a denser motional mode spectrum? In addition, authors now write: “As demonstrated in our experiment, we do not observe any obvious degradation of the simulation when using two modes”. I acknowledge that for a single ion, modes are well separated, but this separation substantially reduces with increasing number of ions. Further, I assume that following the approach, one needs to ensure that ion crystals remain in linear order, which will eventually require that axial frequencies need to be reduced and cooling becomes less efficient, heating rates higher, and the mode-spectrum density increases even further. How does the experimental effort scales with the number of constituent fermions and bosons? The authors need to make a fair statement about these limitations, although these limits may not be of fundamental character.

We thank the Reviewer for his/her interest in the deep technical details of our setup, as well as in our plan to scale the number of controllable ions. In our group we have a clear experimental program for scaling up the system with further increased numbers of fermionic or bosonic modes by trapping and controlling dozens of ions in a single linear trap. When one considers 16 ions in an experiment, using one of the radial motional modes as an example, we may set the trap frequency to about 5 MHz without any serious difficulty. Then, by adjusting the axial trap frequency, we can make the frequencies of the radial modes more or less uniformly distributed in the range of 1 MHz to 5 MHz. In such regime of trap parameters, the frequency separations of the radial modes are around 250 kHz, which are large enough to avoid cross-talks between modes. Comparing to ~500 kHz mode separation that we used in our experimental demonstration, we may need to reduce the laser power by half, preserving the ratio

of coupling to mode frequency separation, in order to have a similar level of cross-talks. We believe that we can push in such a way the number of ions up to a couple of dozen.

In the case of a higher number of ions, where the individual modes cannot be properly addressed, we may pursue the proposal of Casanova et al., Phys. Rev. Lett. 107, 260501 (2011), Ref. [23] in the manuscript. In Ref. [23], it was stated that "Here, it would be advantageous to make use of the transverse modes of a large ion string [22] (Ref. [7] in the manuscript). These modes can be closely spaced in frequency, such that it is not necessary to generate laser frequencies for each of them, relaxing infrastructural requirements while approximating the continuum to a certain extent". Therefore, a possible way to scale to many bosonic modes without the need of individually resolving the modes is to use transverse modes, in a large linear chain, such that the transverse modes are closely spaced and many of them can be addressed by a single bichromatic laser. Thus, the small detuning of each of these modes with respect to the laser frequency plays the role of the free energy of the corresponding mode, and a single laser suffices to generate all the couplings between a given fermionic mode and all the bosonic modes.

We believe these points should be enough to convince the Reviewer. However, we do not want to discuss such further details in the present manuscript, since this is outside the scope of the current manuscript, and will be the subject of future research.

Concerning the description of the "exact dynamics" calculation: Authors write that "We note that there is no free parameter except the initial condition of $|\text{ket}\{0_f 0_f\}$ and the ground state of modes". What do the authors consider to be free parameters? The number of modes, coupling strength, detunings, etc. are also relevant parameters! Did the authors set values to all of these parameters according to experimentally determined parameters, or did they make any kind of numerical model fit to the data? How do experimental parameter uncertainties translate into numerical confidence intervals?

As the Reviewer stated, the number of modes, coupling strengths, detunings, and the like, are relevant parameters. What we mean by 'no free parameters' is that we measure such parameters experimentally and use the values in the simulation. We did not obtain the parameters via fitting. Given that even the initial condition was also determined by experiments, we revised the paragraph as follows,

"We point out that all the parameters in the simulation are experimentally determined, not obtained via fitting."

The authors still write in the main text: "In our simple situation of single bosonic, fermion and anti-fermion modes, we are able to numerically calculate the exact evolution with the full Hamiltonian. For the case that the exact numerical methods are not allowed as the system size grows [...]." And, they did not reply to the earlier questions: "How do the authors define 'not allowed'? At what size or parameter regimes do numerical calculations become unable to verify experimental results due to significant inaccuracy? What kind of formal quality factor did the authors consider in this case? Note, all of these questions need to be carefully clarified, in order to enable a fair judgment of the scientific quality of the results.

When we discuss “not allowed by classical computation”, we are mainly considering the limitations set by the dimension explosion of the Hilbert space for increasing number of constituents, as mentioned in the Discussion. Currently, the full classical simulation via exact diagonalization of a system with about 50 qubits is known to be intractable. Moreover, especially systems with fermionic modes for long range couplings or in more than one spatial dimension are known to have sign problem and therefore are intractable with, e.g., quantum Monte Carlo. This is the situation we will deal with when we scale up our setup. Therefore, our quantum simulation techniques may be advantageous to analyze complex systems of fermions coupled to bosons. In terms of the size of the Hilbert space, our system would outperform numerical simulation as mentioned in the following paragraph of the manuscript,

“In particular, we remark that already with 16 two-level ions and 8 phononic levels per ion, one could perform quantum simulations of interacting quantum field modes that are beyond the reach of classical computations, that is, a Hilbert space dimension of $16^{16} \sim 2^{64}$, which would otherwise require a lengthy quantum algorithm with 64 qubits. This experiment opens an avenue that aims at outperforming the limitations of classical computers, with in principle scalable quantum simulations.”

In the revised version, we updated the corresponding numbers of ions and levels to more clearly show the outperformance of our system beyond the classical limit of computation. We also revised the sentence of the article reading “In our simple situation of single bosonic, fermion and anti-fermion modes, we are able to numerically calculate the exact evolution with the full Hamiltonian. For the case that the exact numerical methods are not allowed as the system size grows [...]” as follows to avoid further confusions:

“In our simple situation of single bosonic, fermion and anti-fermion modes, we are able to numerically calculate the exact evolution with the full Hamiltonian and find a perturbation method that works for a short time dynamics. The whole evolution is then computed by accumulation of the latter. We note that such numerical methods would not be allowed as the system size grows. Typically, one considers the size corresponding to 50 qubits to be intractable. For example, a realistic situation with 16 ions, 16 modes, and 8 considered levels per mode would be beyond the capabilities of classical computers.”

REVIEWERS' COMMENTS:

Reviewer #2 (Remarks to the Author):

November, 7th 2017

In this third revision of the manuscript of X. Zhang et al, entitled "Fermion-antifermion scattering via boson exchange in a trapped ion", authors corrected mistakes and address my previous concerns.

In particular, the discussion of results in Fig. 3 has been made more reasonable and descriptions are now in agreement with the presented figure(s) and captions. I acknowledge that authors refrain from presenting more insights of the experimental scalability prospects in their manuscript, although I suggested otherwise. Unfortunately, the manuscript still exhibits some minor careless mistakes. Moreover, authors present some new statements about numerical tractability that, in my opinion, need clarification and/or references.

To conclude, my final report, I can give only limited support for publication.

Response to referee #2.

Comments: In this third revision of the manuscript of X. Zhang et al, entitled “Fermion-antifermion scattering via boson exchange in a trapped ion”, authors corrected mistakes and address my previous concerns.

In particular, the discussion of results in Fig. 3 has been made more reasonable and descriptions are now in agreement with the presented figure(s) and captions.

Reply: We greatly appreciate your comments that lead us to improve the manuscript significantly.

Comments: I acknowledge that authors refrain from presenting more insights of the experimental scalability prospects in their manuscript, although I suggested otherwise.

Reply: We’ve learned that Nature Communications has a policy on transparent peer review and our discussions of experimental scalability with the reviewer would be open to the public.

Comments: Unfortunately, the manuscript still exhibits some minor careless mistakes. Moreover, authors present some new statements about numerical tractability that, in my opinion, need clarification and/or references.

Reply: We believe we address the mistakes more carefully. In the revised version, we include new references [44,45] related to the numerical tractability in classical computation.